# Process vs. Outcome Reward: Which is Better for Agentic RAG Reinforcement Learning

**Wenlin Zhang**[1,*], **Xiangyang Li**[2,*], **Kuicai Dong**[2,*], **Yichao Wang**[2,†], **Pengyue Jia**[1],
**Xiaopeng Li**[1], **Yingyi Zhang**[1], **Derong Xu**[1], **Zhaocheng Du**[2], **Huifeng Guo**[2],
**Ruiming Tang**[2], **Xiangyu Zhao**[1,†]

[1] City University of Hong Kong, Hong Kong, [2] Noah's Ark Lab, Huawei, China
{wl.z, jia.pengyue, xiaopli2-c, yzhang6375-c, derongxu2-c}@my.cityu.edu.hk,
{lixiangyang34, dong.kuicai, wangyichao5, zhaochengdu, huifeng.guo, tangruiming}@huawei.com,
xianzhao@cityu.edu.hk

## Abstract

Retrieval-augmented generation (RAG) enhances the text generation capabilities of large language models (LLMs) by integrating external knowledge and up-to-date information. However, traditional RAG systems are limited by static workflows and lack the adaptability required for multistep reasoning and complex task management. To address these limitations, agentic RAG systems (e.g., DeepResearch) have been proposed, enabling dynamic retrieval strategies, iterative context refinement, and adaptive workflows for handling complex search queries beyond the capabilities of conventional RAG. Recent advances, such as Search-R1, have demonstrated promising gains using outcome-based reinforcement learning, where the correctness of the final answer serves as the reward signal. Nevertheless, such outcome-supervised agentic RAG methods face challenges including low exploration efficiency, gradient conflict, and sparse reward signals. To overcome these challenges, we propose to utilize fine-grained, process-level rewards to improve training stability, reduce computational costs, and enhance efficiency. Specifically, we introduce a novel method `ReasonRAG` that automatically constructs `RAG-ProGuide`, a high-quality dataset providing process-level rewards for (i) query generation, (ii) evidence extraction, and (iii) answer generation, thereby enhancing model inherent capabilities via process-supervised reinforcement learning. With the process-level policy optimization, the proposed framework empowers LLMs to autonomously invoke search, generate queries, extract relevant evidence, and produce final answers. Compared to existing approaches such as Search-R1 and traditional RAG systems, `ReasonRAG`, leveraging `RAG-ProGuide`, achieves superior performance on five benchmark datasets using only 5k training instances, significantly fewer than the 90k training instances required by Search-R1. Our code is available at https://github.com/Applied-Machine-Learning-Lab/ReasonRAG.

## 1 Introduction

Large language models (LLMs) [1, 2, 3] have demonstrated substantial proficiency in text generation and natural language understanding [4, 5], revealing their potential for powering various downstream applications such as recommender systems [6]. However, their reliance on static training data constrains their ability to address dynamic and real-time queries, often resulting in outdated or hallucinated information [7, 8, 9]. Retrieval-Augmented Generation (RAG) [10] has emerged as a

---

*Equal contribution
†Corresponding Author

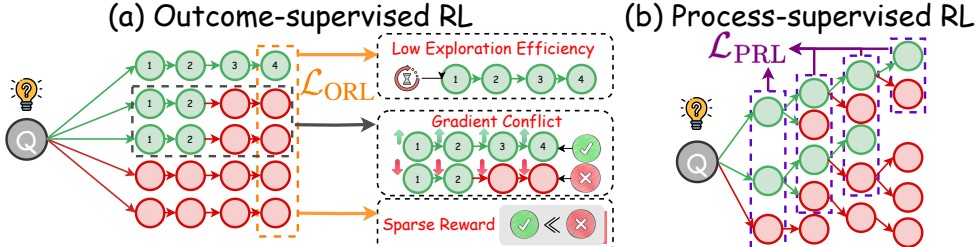

Figure 1: Outcome-supervised vs. process-supervised RL for multi-step reasoning. Each circle denotes one-step reasoning response, where the correct response is colored as "green" and error response is colored as "red".

promising solution by equipping LLMs with external knowledge sources, improving the relevance, factual accuracy, and timeliness of responses [11, 12, 13]. Despite these advances, traditional RAG architectures are limited by their linear and static workflows, which suffer from complex multi-step reasoning, deep contextual integration, and iterative response refinement [10, 14]. To address these shortcomings, agentic RAG (*e.g.,* DeepResearch [15, 16, 17]) systems have been developed, enabling dynamic retrieval strategies, enhanced contextual understanding, and iterative refinement. Achieving agentic RAG requires the underlying LLMs to orchestrate retrieval, filter relevant information, and iteratively refine their outputs, resulting in more adaptive and efficient information processing.

To advance agentic RAG, early approaches [18, 19, 20, 21, 22] primarily focused on prompt engineering to adapt powerful LLMs to agentic workflows. However, due to the limited reasoning and instruction-following capabilities of LLMs, supervised fine-tuning (SFT) methods [23] have been introduced, extending prompt-based approaches by directly optimizing and refining model parameters. Due to SFT storing reasoning steps within the model parameters, the improved reasoning capabilities often encounter challenges in generalizing across different domains [24]. More recently, reinforcement learning (RL) methods (*e.g.,* OpenAI-O1 [25] and DeepSeek-R1 [26] achieve notable improvements in LLM reasoning by employing outcome-supervised RL techniques. Building on these developments, Search-R1 [27] incorporates a search engine as part of the LLM's environment and leverages outcome-based reinforcement learning, using the correctness of the final answer as the reward signal. These advances demonstrate that outcome-supervised reinforcement learning can substantially enhance the capabilities required for agentic RAG, enabling straightforward, end-to-end optimization of the entire workflow.

Despite its promise, outcome-supervised RL also presents inherent limitations, as illustrated in Figure 1. First, **low exploration efficiency** occurs since the model must generate a complete reasoning chain before receiving any reward [27]. Ideally, the reward should be given when errors occur at intermediate steps to facilitate learning. Second, **gradient conflict** arises when mistakes occur late in the reasoning process; the entire sequence (including correct early steps) is penalized [28]. This can lead to conflicting gradients that can push correct actions in the wrong direction. Third, **the rewards are sparse**, as outcome-supervised RL only provides feedback upon producing the final answer [29]. Reward sparsity relies on more training data and steps to converge, as the model receives infrequent learning signals. In contrast, **process-supervised RL** addresses these issues by providing fine-grained, stepwise rewards throughout the reasoning process, enabling more efficient exploration, reducing gradient conflict, and accelerating model learning through denser feedback.

However, applying process-supervised RL to RAG presents several key challenges. (1) **Process Reward Design**: Effective process rewards are essential for guiding the model toward the shortest and most efficient correct reasoning path. Rewards must incentivize helpful intermediate steps that lead to the correct final answer, while penalizing unnecessarily long or circuitous reasoning sequences [30]. (2) **Exploration Efficiency and Annotation Cost**: While human annotators who are skilled in information retrieval can create high-quality process-level annotations by decomposing complex retrieval tasks into efficient steps, this approach is prohibitively expensive due to the substantial manual effort involved [31]. In contrast, autonomous RAG agents can generate a wide range of possible retrieval and reasoning steps, but this large search space makes it difficult to identify and select high-quality, meaningful steps for use as process-level annotations.

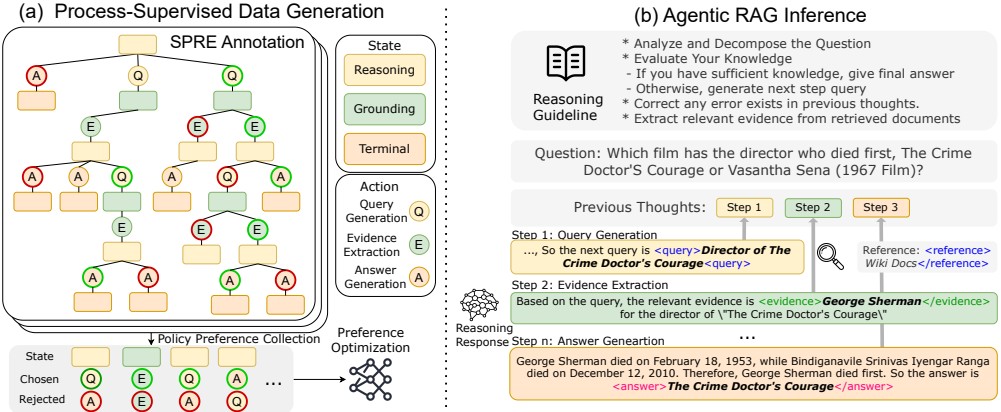

Figure 2: Framework of `ReasonRAG`. Figure (a) illustrates the policy optimization based on process supervision. MCTS guides the construction of the state-action tree and the assignment of process-level rewards for fine-grained policy optimization. (Actions derived from the same state are color-coded by reward: green circle (highest), red circle (lowest).) Figure (b) demonstrates an inference example.

To address these challenges, we propose `ReasonRAG`, an advanced process-supervised RL method to enhance agentic RAG reasoning. Specifically, `ReasonRAG` employs Monte Carlo Tree Search (MCTS) [32] as a search strategy to efficiently balance exploration and exploitation, enabling thorough exploration of diverse reasoning paths and identification of high-reward intermediate steps for guiding the RAG process. Building on these paths, we introduce a novel **S**hortest **P**ath **R**eward **E**stimation (SPRE) algorithm to assign rewards. SPRE favors sequences that lead to the correct answer while penalizing unnecessarily lengthy reasoning, thereby promoting efficiency. This approach yields `RAG-ProGuide`, a dataset comprising 5k queries with 13,000 high-quality process-level preference pairs. Using `RAG-ProGuide` and our process-supervised Direct Preference Optimization (DPO) [33] strategy, `ReasonRAG` is further trained to make dynamic decisions, such as whether to invoke retrieval, formulate subsequent search queries, analyze retrieved documents for relevant evidence, and synthesize evidence into final answers. Extensive experiments on five benchmark RAG datasets show `ReasonRAG` (trained with only 13k process-level steps) outperforms Search-R1 (trained on 90k queries with approximately 270k intermediate steps), suggesting the superiority of process-supervised RL over outcome-supervised RL. Our key contributions can be summarized as follows:

- We propose `ReasonRAG`, an automatic framework for agentic RAG process-level reward annotations. We introduce SPRE for efficient RAG process-level reward annotation and MCTS for high-quality decision space exploration.
- We introduce a process-level annotation dataset `RAG-ProGuide`, which serves as an off-policy dataset, and can be easily applied for various LLM policy optimization.
- We conduct extensive comparative experiments of outcome-supervised RL and process-supervised RL for RAG reasoning with Qwen2.5-7B-Instruct. The experimental results on five benchmark datasets demonstrate the superiority and training efficiency of `ReasonRAG`.

## 2 ReasonRAG Framework

### 2.1 Framework Overview

This section details the design of `ReasonRAG` framework, as depicted in Figure 2. Figure 2a outlines our approach for constructing high-quality process-supervised data. We first introduce *Shortest Path Reward Estimation* (SPRE) to provide process-level supervision reward (see Section 2.2.1). To efficiently gather these rewards, we employ Monte Carlo Tree Search (MCTS) algorithm to explore the vast decision space in agentic RAG and collect informative intermediate steps (see Section 2.2.2).

The resulting process-supervised dataset, `RAG-ProGuide` (see Section 2.2.3), is then used to optimize `ReasonRAG` via *policy preference optimization*. This training strategy guides the model to prefer desirable reasoning trajectories in agentic RAG (see Section 2.3).

Figure 2b illustrates the agentic RAG inference workflow in `ReasonRAG`. During inference, the model adaptively conducts reasoning by dynamically invoking search engine and interleaving three core actions: query generation, evidence extraction, and answer generation (see Section 2.4).

## 2.2 Process-Supervised Data Generation

Effective process-supervised policy optimization requires high-quality, granular reward signals at the process level. As outlined in Section 1, generating such rewards for agentic RAG presents two main challenges: (1) the absence of reward functions for intermediate reasoning steps, and (2) the lack of an efficient and cost-effective method to generate informative reasoning trajectories. To overcome these challenges, we introduce a novel process-level reward function, SPRE (see Section 2.2.1), specifically designed for agentic RAG. Furthermore, we develop an MCTS-based approach (see Section 2.2.2) to efficiently explore the decision space and collect high-quality process-level data.

### 2.2.1 Shortest Path Reward Estimation (SPRE)

Unlike outcome-level rewards, process-level rewards provide supervision at each intermediate step of agentic RAG. A key challenge is the absence of ground-truth reward signals for partial reasoning trajectories. Furthermore, due to the large decision space, the reward function must account for both final correctness and reasoning efficiency. To address these challenges, we propose *Shortest Path Reward Estimation (SPRE)*, which evaluates the quality of each intermediate reasoning path by simulating its possible outcomes and penalizing unnecessarily long trajectories.

Formally, the agentic RAG process consists of an $n$-step sequence $[y_1, \cdots, y_n]$, where each $y_i$ represents the output of a single reasoning step, conditioned on the initial question $x$ and previous steps $y_{<i}$. To evaluate a partial sequence $y_{1:t}$, we simulate multiple continuations, known as *rollouts*, until a final answer is obtained. By repeating the rollout process $k$ times and scoring each outcome, we approximate the reward as a Monte Carlo-style estimation with step-based penalties:

$$Q_t = \text{MonteCarlo}(x, y_{1:t}) = \frac{1}{k} \sum_{i=1}^{k} v(\text{rollout}_i) \cdot \alpha^{\text{step}(\text{rollout}_i)} \tag{1}$$

Here, rollout$_i$ is the $i$-th simulated completion of $y_{1:t}$, $v(\text{rollout}_i) \in [0, 1]$ denotes the correctness score (*e.g.,* $F_1$ match to the ground truth), and step(rollout$_i$) is the number of total reasoning steps in the trajectory. The decay factor $\alpha \in (0, 1]$ penalizes unnecessarily long reasoning paths. This reward encourages the model to favor trajectories that achieve correct answers with fewer steps, thus balancing accuracy and efficiency in agentic RAG reasoning.

### 2.2.2 Monte Carlo Tree Search (MCTS) for Process-level Exploration

Although SPRE offers reliable reward signals for evaluating intermediate steps, generating diverse yet meaningful trajectories remains challenging. The search space in agentic RAG is extensive due to open-ended nature of retrieval, which requires continuous refinement of search queries for relevant information. To address this, we propose a tailored MCTS framework for agentic RAG. MCTS enables efficient exploration by selectively expanding the most promising reasoning paths based on estimated rewards.

We adapt MCTS to agentic RAG context by explicitly defining states and actions for tree construction. Formally, each intermediate reasoning step is represented as a state $s = (x, y_{<i}, \text{stage})$, where $x$ is the original question, $y_{<i}$ is the sequence of prior reasoning outputs, and stage $\in$ {Reasoning, Grounding, Terminal} indicates current point of agentic flow. Actions $\in$ {Query Generation, Evidence Extraction, Answer Generation} are determined by the current stage as follows:

- **Reasoning** stage: Choose between generating a new query for document retrieval or directly generating an answer. If a new query is generated, a retrieval operation is performed, and the retrieved documents are appended to the state for subsequent decisions. If an answer is produced, the process transitions to the *Terminal* stage.
- **Grounding** stage: Select evidence spans from the retrieved documents. Afterwards, the system returns to the *Reasoning* stage for further reasoning or answering.
- **Terminal** stage: End the exploration process when the final answer has been generated.

| Statistics | Number |
|---|---|
| **Questions** | 4603 |
| - PopQA | 704 (15.3%) |
| - HotpotQA | 2843 (61.8%) |
| - 2WikiMultihopQA | 1056 (22.9%) |
| **Actions** | 13289 |
| - Query Generation | 3295 (24.8%) |
| - Evidence Extraction | 4305 (32.4%) |
| - Answer Generation | 5689 (42.8%) |
| Avg./Min./Med./Max. Iteration | 2.7/1/3/5 |
| Avg./Min./Med./Max. Tokens | 65.5/9/60/625 |

Table 1: Overall Dataset Statistics

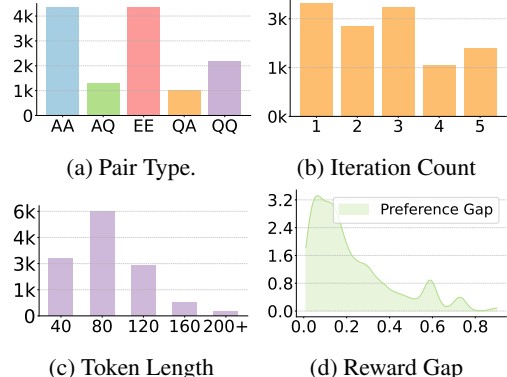

(a) Pair Type.   (b) Iteration Count

(c) Token Length   (d) Reward Gap

Figure 3: Dataset Distribution.

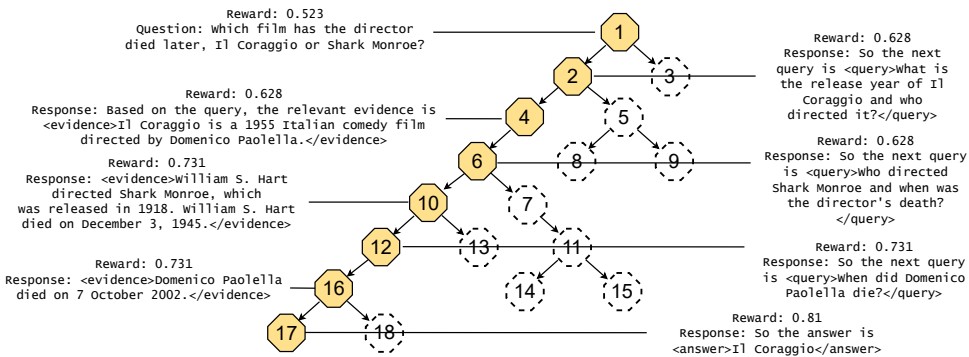

Figure 4: A Tree Example for the process-level preference annotations.

Based on the current state $s$, the policy for generating the next action $a$ is defined as:

$$\pi(a \mid s) = \text{LLM}(a \mid s) = \begin{cases} \pi_\theta(\cdot \mid x, y_{<i}, p_{\text{stage}}), & \text{if stage is Reasoning} \\ \pi_\theta(\cdot \mid x, y_{<i}, \text{docs}, p_{\text{stage}}), & \text{otherwise} \end{cases} \quad (2)$$

State transitions are defined as $s_{t+1} = \text{concatenate}(s_t, a_t)$, where each action leads to a new node in the search tree, representing an extended reasoning sequence. This recursive process incrementally builds a tree rooted at the original question. MCTS then operates iteratively, performing three core steps: *selection*, *expansion*, and *backpropagation*. Specifically, at each iteration, MCTS selects promising paths using a Upper Confidence Bound (UCB) based objective, expands new states by sampling LLM-generated actions, and backpropagates SPRE-estimated rewards (see Equation 1) to update the tree (see more comprehensive explanation of MCTS exploration in Appendix A). Integrating MCTS with SPRE enables efficient exploration and prioritization of high-reward reasoning steps, producing high-quality process-level annotations to optimize the agentic RAG policy.

### 2.2.3 `RAG-ProGuide` **Dataset**

**Construction.** Using the MCTS-based exploration framework, we construct a high-quality process-supervised dataset `RAG-ProGuide` to facilitate process-level policy optimization. We randomly sample 3,000 questions each from PopQA [34], HotpotQA [35], and 2WikiMultihopQA [36], covering both single-hop and multi-hop question answering tasks. An advanced large language model serves as the policy model to simulate the agentic RAG reasoning process within the MCTS framework (see Section 2.2.2). During tree search, we prune all branches that do not yield a final answer. For each complete trajectory, we compute the $F_1$ score between the predicted answer and ground truth, and use this correctness signal to estimate intermediate node rewards via SPRE (see Section 2.2.1). These rewards are propagated through the MCTS tree to guide preference pair selection. To ensure high-quality preference data, we perform post-processing to remove duplicates and uninformative

comparisons: (i) we discard preference pairs with identical response sequences, and (ii) pairs with a reward difference less than 0.01. After filtering, the final dataset consists of 4,603 questions and 13,289 distinct preference pairs. Figure 4 illustrates an example for the tree-structured process data. The root node corresponding to the original question, each node corresponds one-step response from LLMs. The correct reasoning path has been colored in orange and annotated with a higher reward.

**Dataset Statistics.** Table 1 presents detailed statistics and distributions for our dataset. The questions are drawn from PopQA, HotpotQA, and 2WikiMultihopQA, providing comprehensive coverage of both single-hop and multi-hop reasoning within the RAG decision space. The dataset contains a balanced distribution of three reasoning actions, reflecting the multi-stage nature of the agentic RAG process. As shown in Figure 2a, the distribution of preference pair types demonstrates diverse comparative scenarios; the x-axis abbreviations (**A**: answer generation, **Q**: query generation, **E**: evidence extraction) indicate action types in accepted versus rejected paths. This diversity ensures fine-grained comparative coverage across different reasoning stages. Figure 2b indicates a range of reasoning iteration counts, consistent with the complexity of multi-hop inference. Figure 2c shows a broad distribution of response token lengths, confirming the dataset's capacity to capture various response complexities. Additionally, Figure 2d illustrates the probability distribution of the reward gap between preference pairs. The comprehensive coverage of this distribution across a wide range of values ensures that the preference learning is guided by a rich and informative signal. Collectively, these statistics demonstrate the dataset's quality and its suitability for training robust process-level decision policies in agentic RAG frameworks.

## 2.3 Process-Supervised Preference Optimization

Based on the process-supervised preference data, we apply DPO [33] to tune the policy model. The optimization objective can be denoted as follows:

$$\mathcal{L}(\theta) = -\mathbb{E}_{(x, y_{<t}, y_t^w, y_t^l) \sim \mathcal{D}} \left[ \log \sigma \left( \beta \log \frac{p_\theta(y_t^w | x, y_{<t})}{p_\theta(y_t^l | x, y_{<t})} \right) \right] \tag{3}$$

where $x$ denotes the original question, $y_{<t}$ denote the the responses from previous reasoning steps, $y_t^w$ and $y_t^l$ represent the preferred and dispreferred responses in the subsequent step, respectively, and the hyperparameter $\beta$ controls the KL constraint.

## 2.4 Agentic RAG Inference

To enable LLMs to autonomously interact with external information, we propose an agentic RAG workflow that supports adaptive reasoning through iterative search and reflection. `ReasonRAG` allows the model to dynamically determine when and how to invoke search engine based on question complexity.

The workflow operates through three recursive decision states: Reasoning, Grounding, and Terminal. In the **Reasoning** state, the LLM evaluates the current context to decide if it has sufficient information to answer the question. If sufficient, it generates a final answer enclosed in placeholders (`<answer>` and `</answer>`), thus terminating the process. If not, the model creates a new query enclosed in `<query>` and `</query>` placeholders to retrieve additional evidence. The system then transitions to the **Grounding** state, where documents are retrieved based on the query, and the model extracts relevant evidence spans. These evidence spans are appended to the context, after which the process loops back to the **Reasoning** state for further deliberation. (See Appendix F for detailed prompt designs.)

---

**Algorithm 1** Agentic RAG Inference Pipeline

---

**Require:** Original question $x$, large language model $\pi_\theta$, retriever $\mathcal{R}$, maximum reasoning round $N$.
**Ensure:** Final response $y$.

1: Initialize reasoning count $i \leftarrow 0$, and stage $\leftarrow$ Reasoning
2: **for** $i \leftarrow 0$ **to** $N - 1$ **do**
3:     **if** stage is Reasoning **then**
4:         $y_i \sim \pi_\theta(\cdot | x, y_{<i}, p_{\text{stage}})$
5:     **else**
6:         $y_i \sim \pi_\theta(\cdot | x, y_{<i}, docs, p_{\text{stage}})$
7:     **end if**
8:     $y \leftarrow y + y_i$
9:     **if** <query> detected in $y_i$ **then**
10:         stage $\leftarrow$ Grounding
11:         $q \leftarrow$ extract_query($y_i$)
12:         docs $\leftarrow \mathcal{R}(q)$
13:     **else if** <answer> detected in $y_i$ **then**
14:         stage $\leftarrow$ Terminal
15:         **return** extract_answer($y_i$)
16:     **else if** <evidence> detected in $y_i$ **then**
17:         stage $\leftarrow$ Reasoning
18:     **end if**
19: **end for**
20: **return** final response $y$

---

Table 2: Main Results (%) on Five benchmarks (the number of queries used for training is indicated in brackets). "*" indicates the statistically significance (i.e., two-sided t-test with $p < 0.05$) over the best baseline. Two most important columns: the averaged EM and $F_1$ are highlighted.

| Type | Method | PopQA | | HotpotQA | | 2WikiMulti | | Bamboogle | | MuSiQue | | Avg. | |
|------|--------|------|------|------|------|------|------|------|------|------|------|------|------|
| | | EM | $F_1$ | EM | $F_1$ | EM | $F_1$ | EM | $F_1$ | EM | $F_1$ | EM | $F_1$ |
| Zero-shot | Naïve Generation | 12.7 | 16.5 | 15.7 | 24.8 | 20.2 | 28.0 | 6.4 | 17.4 | 2.7 | 10.2 | 11.5 | 19.4 |
| | Standard RAG | 38.4 | 44.7 | 29.3 | 39.9 | 29.4 | 36.3 | 17.6 | 24.1 | 6.7 | 15.1 | 24.3 | 32.0 |
| Active | FLARE | 14.3 | 17.6 | 18.1 | 25.7 | 27.9 | 32.8 | 12.0 | 20.8 | 4.3 | 12.6 | 15.3 | 21.9 |
| | Self-RAG(146k) | 22.7 | 33.9 | 21.0 | 29.7 | 12.0 | 25.2 | 1.6 | 10.9 | 4.6 | 13.3 | 12.4 | 22.6 |
| Adaptive | AdaptiveRAG(3k) | 36.6 | 41.5 | 29.1 | 40.7 | 24.2 | 33.4 | 18.4 | 26.1 | 6.9 | 14.3 | 23.0 | 31.2 |
| RAG-CoT | Iter-Retgen | 38.7 | 44.9 | 30.3 | 42.1 | 31.2 | 38.7 | 19.2 | 26.4 | 7.7 | 14.2 | 25.4 | 33.3 |
| | IRCoT | 36.2 | 43.6 | 27.7 | 41.5 | 23.5 | 32.5 | 17.2 | 22.5 | 8.6 | 13.2 | 22.6 | 30.7 |
| Summary | RECOMP | 40.5 | 45.8 | 29.7 | 41.2 | 33.2 | 39.4 | 21.7 | 28.6 | 9.2 | 15.8 | 26.9 | 34.2 |
| | LongLLMLingua | 39.2 | 45.1 | 31.4 | 43.2 | 34.5 | 40.2 | 20.3 | 27.4 | 8.7 | 14.9 | 26.8 | 34.2 |
| | Selective-Context | 34.9 | 41.5 | 19.3 | 27.3 | 20.3 | 29.7 | 15.3 | 22.6 | 6.1 | 13.7 | 19.2 | 27.0 |
| Reasoning | Search-o1 | 33.2 | 40.3 | 24.8 | 38.1 | 16.4 | 27.1 | 30.4 | 40.6 | 6.3 | 13.7 | 22.2 | 31.96 |
| | AutoRAG(10k) | 38.6 | 44.1 | 33.3 | 43.7 | 39.5 | 46.1 | 24.8 | 32.2 | 11.3 | 18.3 | 29.5 | 36.9 |
| | Search-R1(90k) | 39.7 | 44.8 | 37.0 | 47.0 | 41.4 | 48.0 | 32.0 | 43.8 | 14.6 | 19.9 | 32.8 | 40.7 |
| | **ReasonRAG(5k)** | **41.5\*** | **46.2\*** | **38.4\*** | **48.9\*** | **43.6\*** | **50.4\*** | **36.0\*** | **45.5\*** | 12.8 | 20.6\* | **34.4\*** | **42.3\*** |

In summary, `ReasonRAG` supports multi-step, flexible reasoning while maintaining structured decision control. The use of explicit placeholders enhances interpretability and facilitates programmatic control during deployment. The complete algorithmic flow is provided in Algorithm 1.

# 3 Experiments

## 3.1 Experimental Setup

**Evaluation Dataset & Metrics.** We evaluate `ReasonRAG` and all baselines on five public benchmarks: the single-hop QA dataset PopQA [34] and four multi-hop QA datasets, including HotpotQA [35], 2WikiMultiHopQA [36], Bamboogle [37], and MuSiQue [38]. Bamboogle and MuSiQue serve as out-of-domain QA evaluation datasets. The diversity of these datasets enables a comprehensive assessment of agentic RAG. We report Exact Match (EM) and $F_1$ scores as evaluation metrics. Refer to Appendix D for more details about dataset introduction, statistics, and metrics.

**Baselines.** We implement 12 baseline models which can be categorized into 6 types as follows: **Zero-shot:** Directly use prompt engineering on LLM to answer the question without or with retrieved documents [39]. **Active:** Actively make additional retrieval when retrieved data or generated responses have low confidence [40, 41]. **Adaptive:** Dynamically chooses the most suitable RAG pipeline from no-retrieval, single-hop, or multi-hop retrieval strategies [42]. **RAG-CoT:** Integrates chain-of-thought reasoning with retrieval, enabling multi-step, evidence-seeking answers [43, 44]. **Summary:** Compresses or summarizes retrieved content to fit model input constraints while retaining key information [45, 46, 47]. **Reasoning:** Enhances multi-hop reasoning by structuring the reasoning process and scrutinizing retrieved evidence [48, 27, 49]. Note that `ReasonRAG` and all baselines use Qwen2.5-7B-Instruct [50] as the backbone model, ensuring fair comparison. Refer to more implementation details about `ReasonRAG` and baselines in Appendix E.1 and E.2.

## 3.2 Main Results

We present detailed performance results on `ReasonRAG` against 12 baselines across five benchmark datasets, as shown in Table 2. Our key findings are summarized below:

- **Data Efficiency:** `ReasonRAG`, despite being trained on only 5k queries, outperforms the search-R1 baseline trained with 90k queries. On average across all datasets, `ReasonRAG` achieves higher EM

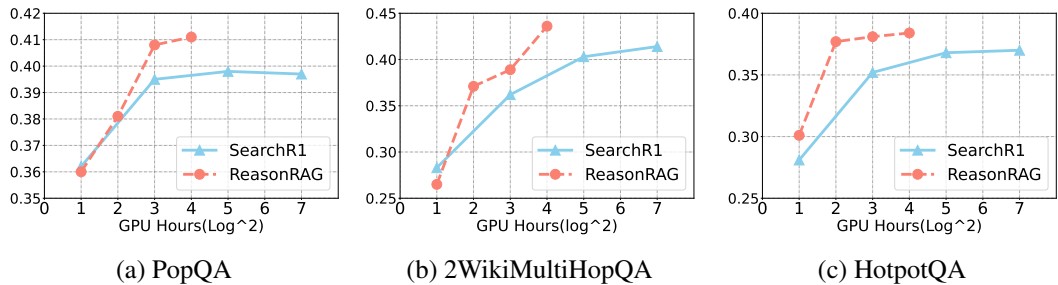

(a) PopQA    (b) 2WikiMultiHopQA    (c) HotpotQA

Figure 5: Training cost and convergence speed comparison (EM%) for ReasonRAG and Search-R1

Table 3: Impact of different optimization strategies on `ReasonRAG` 's effectiveness.

| Method | PopQA | | HotpotQA | | 2WikiMulti | | Bamboogle | | MuSiQue | | Avg. | |
|---|---|---|---|---|---|---|---|---|---|---|---|---|
| | EM | $F_1$ | EM | $F_1$ | EM | $F_1$ | EM | $F_1$ | EM | $F_1$ | EM | $F_1$ |
| ReasonRAG (Base) | 35.6 | 42.7 | 23.7 | 38.2 | 15.2 | 28.9 | 28.0 | 38.7 | 7.7 | 15.4 | 22.0 | 32.8 |
| ReasonRAG (SFT) | 31.6 | 37.4 | 26.8 | 38.7 | 35.1 | 40.9 | 17.6 | 27.3 | 8.6 | 15.5 | 23.9 | 32.0 |
| ReasonRAG (RL-ORL): 5k queries | 23.0 | 30.9 | 28.1 | 32.6 | 32.0 | 43.8 | 17.5 | 24.1 | 5.9 | 13.1 | 21.3 | 28.9 |
| ReasonRAG (RL-ORL): 10k queries | 39.5 | 45.7 | 36.7 | 46.7 | 40.5 | 47.2 | 30.7 | 40.6 | 12.6 | 19.5 | 32.0 | 39.9 |
| ReasonRAG (RL-PRL) | 41.5 | 46.2 | 38.4 | 48.9 | 43.6 | 50.4 | 36.0 | 45.5 | 12.8 | 20.6 | 34.5 | 42.3 |

(34.4%) and $F_1$ (42.3%) scores compared to search-R1 (32.8% EM, 40.7% $F_1$), highlighting the superior data efficiency of `ReasonRAG`. This demonstrates the effectiveness of process-supervised RL, which leverages fine-grained rewards, over current outcome-supervised methods.

- **Multi-hop Reasoning:** `ReasonRAG` shows substantial performance gains on multi-hop reasoning tasks. On the HotpotQA dataset, it achieves an $F_1$ score of 48.9%, outperforming models like AutoRAG (43.7% $F_1$) and search-R1 (47.0% $F_1$), both of which are trained on larger datasets. This underscores `ReasonRAG` 's strength in handling complex, multi-step questions that require integrating evidence from multiple sources.

- **Out-of-domain Generalization:** `ReasonRAG` demonstrates strong generalization to out-of-domain data. On challenging benchmarks such as Bamboogle and MuSiQue, it consistently achieves higher $F_1$ scores relative to other baselines. This indicates improved robustness and transferability of its reasoning capabilities across different domains.

### 3.3 Training Efficiency

Figure 5 compares the training efficiency of `ReasonRAG` and Search-R1. The figure illustrates the progression of EM scores with increasing GPU hours across three datasets. The results reveal that `ReasonRAG` has higher training efficiency compared to Search-R1. `ReasonRAG` achieves superior EM scores with fewer GPU hours, indicating that it requires less training data and compute to reach strong performance levels. In contrast, Search-R1 requires significantly more GPU hours to reach similar performance.

The efficiency gap between the two models is particularly notable on multi-hop question answering tasks. For the single-hop PopQA dataset, performance gains for both models are comparably rapid as training progresses. However, for multi-hop datasets such as 2WikiMultiHopQA and HotpotQA, `ReasonRAG` consistently demonstrates significant improvements with increased GPU hours. This further underscores its effectiveness on complex reasoning tasks, where it delivers faster and greater performance improvements with fewer resources.

### 3.4 Effectiveness of Different Optimization Strategies

In this section, we compare the effectiveness of `ReasonRAG` utilizing three different optimization strategies against the base model. Our default approach, `ReasonRAG` (RL-PRL), is trained with process-level rewards as described in Section 2. For `ReasonRAG` (RL-ORL), we adopt outcome-level reward training following the Search-R1 protocol [27]. Specifically, we evaluate two versions:

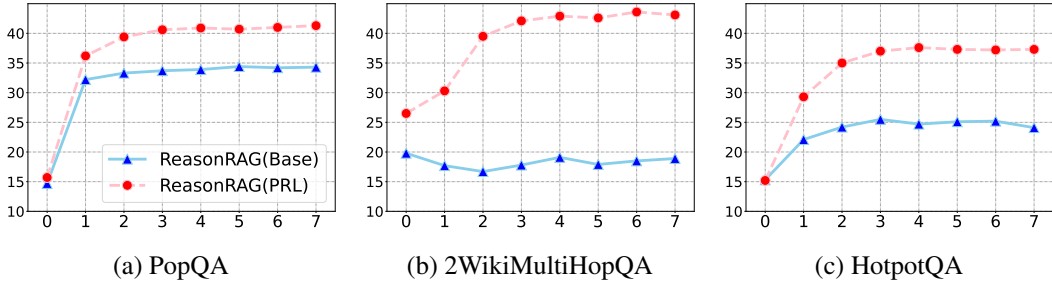

Figure 6: The EM performance across varying retrieval iterations on 3 benchmarks.

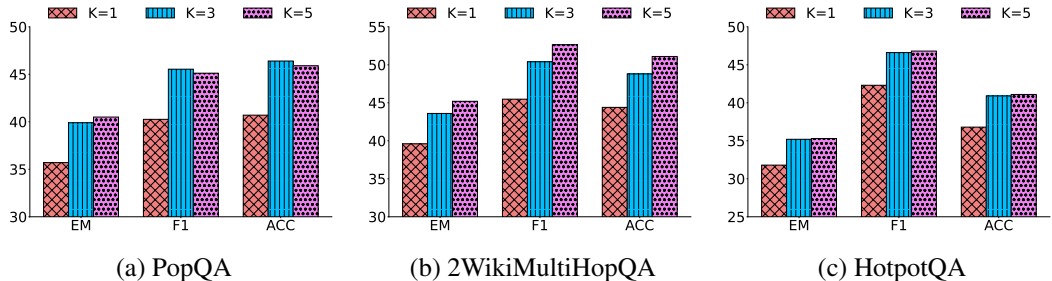

Figure 7: Effect of top-$k$ retrieved documents on `ReasonRAG`'s performance across 3 datasets.

RL-ORL-5k, trained on the same 5k queries as RL-PRL, and RL-ORL-10k, which incorporates an additional 5k queries sampled from PopQA, HotpotQA, and 2WikiMultiHopQA, totaling 10k training examples. For `ReasonRAG` (SFT), we use the preferred responses from the `RAG-ProGuide` preference pairs as ground truth and apply supervised fine-tuning (SFT) via next-token prediction. Table 3 summarizes the performance of these four variants. Our main findings are as follows:

- **Superiority of PRL**: `ReasonRAG` (PRL) consistently outperforms all other variants across all datasets, both in-domain and out-of-domain, indicating stronger generalization capabilities.
- **High Data Demand of ORL**: `ReasonRAG` (ORL) achieves the second-best results, but requires substantially more training data to match the comparable performance of PRL. Although ORL is more effective than Base and SFT, its training efficiency is relatively low.
- **Overfitting in SFT**: SFT leads to overfitting on multi-hop reasoning paths, resulting in reduced performance on single-hop tasks. Furthermore, SFT-trained models generalize poorly, as demonstrated by a marked performance decline on the Bamboogle dataset.

### 3.5 Impact of Search on Performance

**Performance vs. Retrieval Steps.** Figure 6 shows the EM performance of `ReasonRAG` across varying retrieval iterations on 3 datasets. We observe a consistent trend: performance improves with more retrieval iterations and then gradually saturates. Notably, `ReasonRAG` can adaptively determine the required inference depth according to task complexity. For the single-hop PopQA dataset, performance converges within 2 to 3 retrieval steps, whereas more complex multi-hop tasks such as 2WikiMultiHopQA and HotpotQA require 3 to 5 steps to reach peak performance. In contrast, `ReasonRAG` (base) without preference optimization only achieves reliable gains on PopQA and struggles to handle multi-hop settings. These results demonstrate `ReasonRAG` 's ability to perform adaptive reasoning in response to the complexity of the input question.

**Performance vs. Number of Retrieved Documents.** Figure 7 compares the performance of `ReasonRAG` under different top-$k$ retrieval settings, where $k$ refers to retrieving top-$k$ relevant document passages per search query. Results indicate that while `ReasonRAG` remains robust across a range of $k$ values, its performance is sensitive to the quantity of retrieved information. With $k = 1$, limited context restricts the model's reasoning ability. Increasing $k$ to 3 yields significant improvements across all datasets, suggesting that `ReasonRAG` effectively leverages additional evidence. Further increasing $k$ to 5 does not further increase the performance on PopQA and HotpotQA, whereas

2WikiMultiHopQA continues to benefit from richer retrieved context. Overall, these findings highlight `ReasonRAG` 's capacity to utilize additional retrieved documents, particularly in more complex multi-hop scenarios.

## 4 Related Works

**Prompt-Based Agentic RAG.** Early prompt-based approaches leverage manually designed workflows to elicit the inherent capabilities of LLMs for interacting with external information. Specifically, the RAG task is often decomposed into subtasks such as adaptive retrieval judgment [42], query generation [21, 43, 44], evidence extraction [41, 45, 47, 46], and answer generation [40]. While some efforts have focused on optimizing RAG through personalization [51], graph-based retrieval [52], or reranking techniques [53], yet a critical gap remains in designing LLMs that can autonomously invoke search engines. Recently, agentic RAG aims to design workflows that empower LLM to autonomously interact with external information. OpenResearcher [54], AirRAG [55], IterDRAG [56], PlanRAG [57], and Search-o1 [49] demonstrate strong performance improvement by the effective incorporation with the search engine. Nevertheless, these methods are limited by their dependence on inherent capabilities for interacting with external information and the requirement for manual design when applied to new domains, and lack explicit mechanisms for eliminating distracting information [58].

**RL-Based Agentic RAG.** Reinforcement Learning (RL) has consistently delivered significant performance gains across a spectrum of sequential decision-making tasks, as evidenced by its successful application in domains such as recommender systems [59, 60, 61, 62, 63]. Recently, the success of models like DeepSeek-R1 [26] has vividly demonstrated the substantial potential of outcome-supervised RL in enhancing complex reasoning capabilities, establishing it as a mainstream paradigm for end-to-end optimization of LLMs. Following the widespread adoption of RL by major AI companies to improve the reasoning abilities of their models on complex tasks [64, 65, 66], recent work [27, 67, 68, 69] has extended outcome-supervised reinforcement learning to RAG, empowering LLMs to autonomously utilize search engines for intricate inference. While outcome-supervised RL has demonstrated significant performance gains, it also faces challenges such as reward sparsity, training instability, and substantial computational cost. Moreover, outcome-supervised RL typically demands extensive training resources. In contrast, process-supervised RL has recently been applied to enhance reasoning abilities, outperforming outcome-supervised approaches by providing fine-grained rewards [70, 71, 72, 73]. As an alternative avenue for improving LLM reasoning in RAG, process-supervised RL for RAG remains unexplored.

## 5 Conclusion

We introduce `ReasonRAG`, a process-supervised agentic RAG method for fine-grained policy optimization. Our approach integrates Monte Carlo Tree Search (MCTS) with the agentic RAG workflow to generate `RAG-ProGuide`, a high-quality dataset providing process-level supervision by prioritizing the shortest reasoning paths leading to correct answers. Leveraging `RAG-ProGuide`, we perform preference-based policy optimization to enhance LLMs' autonomous capabilities in query generation, evidence extraction, and answer synthesis. Experiments demonstrate that `ReasonRAG` achieves superior performance on five benchmark datasets using only 5k training instances, significantly fewer than the 90k required by Search-R1, highlighting the effectiveness of `RAG-ProGuide` 's high-quality process-level rewards in optimizing agentic RAG policies.

### Acknowledgements

This research was partially supported by National Natural Science Foundation of China (No.62502404), Hong Kong Research Grants Council's Research Impact Fund (No.R1015-23), Research Grants Council's Collaborative Research Fund (No.C1043-24GF), Research Grants Council's General Research Fund (No.11218325), Graduate Research Fund of the School of Economics and Management of Dalian University of Technology (No. DUTSEMDRFKO1), Institute of Digital Medicine of City University of Hong Kong (No.9229503), and Huawei (Huawei Innovation Research Program).

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

## Appendix Overview

The appendix includes the following sections:

- **Appendix A**: Details the MCTS tree construction process using process-level rewards. This serves as an extended explanation to Section 2.2.2.
- **Appendix B**: Illustrates the example of LLM evaluation for process-level simulation. This serves as an extended example of MCTS node expansion in Section 2.2.2.
- **Appendix C**: Compares the reasoning response between ReasonRAG(base) and ReasonRAG(PRL). This serves as an extended example for Section 3.4.
- **Appendix D**: Provides additional details of the evaluation setup. This serves as a supplement to Section 3.1.
- **Appendix E**: Provide more details of the implementation. This serves as a supplement to Section 3.1.
- **Appendix F**: Details the prompt design for Agentic RAG. This serves a supplement to Section 2.4.
- **Appendix G**: Details the licensing terms and conditions governing the use and distribution of the proposed datasets.
- **Appendix H**: Discusses the limitations and constraints of the proposed approach.
- **Appendix I**: Evaluates the potential societal implications and ethical considerations of the research.

## A  Monte Carlo Tree Search for Process-level Reward

Formally, for each RAG intermediate process, its corresponding state $s$ encompasses the original question $x$, the preceding thoughts $y_{<i}$, and a stage $\in \{$Reasoning, Grounding, Terminal$\}$. The stage indicates the current decision mode within the reasoning process. In the Reasoning stage, the LLM autonomously decides whether to generate a new query or directly produce an answer. If a query is chosen, it triggers a call to the external search engine, and the retrieved documents are added to the context in the next step. If the model opts to generate an answer instead, the process transitions into the Terminal stage. In the Grounding stage, the model extracts relevant evidence spans from the retrieved documents based on the most recent query. After extracting evidence, the state transitions back to the Reasoning stage, enabling further iterative reasoning.

Based on the state, the policy of the next action is defined as

$$\pi(a|s) = \mathrm{LLM}(a|s) = \begin{cases} \pi_\theta(\cdot|x, y_{<i}, p_{\text{stage}}), & \text{if stage is Reasoning} \\ \pi_\theta(\cdot|x, y_{<i}, \text{docs}, p_{\text{stage}}), & \text{otherwise} \end{cases} \tag{4}$$

Consequently, the state transition can be represented as $s_{t+1} = \text{concatenate}(s_t, a_t)$. Each node in the tree contains the following information: $\{N(s), Q(s), \text{Stage}(s)\}$, where $N(s)$ denotes the number of times state $s$ has been visited, $Q(s)$ represents the current intermediate annotation collected through the Monte Carlo method, with values in the range $[0, 1]$. With the tree structure defined, the MCTS begins from the root node and constructs the tree by iteratively performing three key operations: selection, expansion, and backpropagation.

**Selection:** This step aims to select nodes that balance the search quality and exploration degree. The node selection starts from the root node, and iteratively selects the child nodes based on their state value $Q$ and visiting frequency $N$. These variable are refined during the search strategy, detailed in the backpropagation section. To effectively trade off between exploring unvisited nodes and exploiting nodes with higher state value, we iteratively search for the next node using UCT score [74]. The state is selected according to the following formula:

$$s_i^* = \text{argmax}_{s_i \in \text{children(s)}}[Q(s_i) + c_{uct}\frac{\sqrt{\sum_i N(s_i)}}{1 + N(s_i)}] \tag{5}$$

where $c_{uct}$ is a trade-off parameter to control the exploration degree. The algorithm begins by exploring unvisited states and progressively favors nodes with higher Q-values and fewer visits.

**Expansion:** Given a selected node that does not reach a terminal state and maximum child node limit, the expansion step proceeds with a single step of RAG reasoning based on Equation ( 4) and initializes a new child node with the generated response. Following response generation, the simulation step iteratively reasons until a final answer is reached, serving as the basis for initializing the reward of the created node. However, simulating RAG leads to inefficiency due to its need for iterative LLM reasoning and retrieval. To address this challenge, the correctness of the intermediate reasoning process is evaluated by LLM judgment based on the intermediate process against the golden answer, outputting a correctness value $v \in [0, 1]$, as defined in Equation (6). This approach avoids time-consuming simulations, providing an efficient evaluation for exploring the complex RAG reasoning space.

$$v(s) = \pi_\theta(\cdot|x, y_{1:t}, \text{answer}, p_{\text{eval}}) \tag{6}$$

**Backpropagation:** Following node expansion, the node statistics are iteratively updated from the current node back to the root node. The correctness score $Q(s_t)$ is updated based on Eq. (1). The visit count for the traversed nodes is incremented:

$$N(s) \leftarrow N(s) + 1 \quad \forall s \in \text{path from } s_t \text{ to root} \tag{7}$$

# B   Process Evaluation

The three cases illustrated in Figures 8–10 demonstrate the effectiveness of `ReasonRAG` in supporting stepwise reasoning over retrieved content. In the partially correct case, the agent retrieves relevant evidence but fails to cover all required entities, leading to incomplete inference. In the incorrect case, the agent misidentifies directors, resulting in a reasoning chain based on factual errors. In contrast, the fully correct case shows `ReasonRAG`'s ability to decompose the question into sequential sub-queries, accurately retrieve biographical information, and synthesize the correct answer through structured comparison. Although minor imperfections exist in the query formulation, the reasoning trace demonstrates clear progression, factual grounding, and answer justification. These cases collectively highlight `ReasonRAG`'s advantage in aligning sub-question formulation with retrieval intent and maintaining semantic coherence across reasoning steps, leading to improved robustness and interpretability in complex multi-hop scenarios.

# C   Case Study

Figure 11 shows an example for agentic RAG response from `ReasonRAG`. Before the policy optimization, LLMs fails to generate the appropriate query and mislead by irrelevant information. In contrast, the process-supervised RL empower LLMs to autonomously invoke query generation, evidence extraction, and answer generation.

## Case of LLM judge partial correct process

**Question**
``Which film has the director died later, Il Coraggio or Shark Monroe?''
**Golden Answers**
``Il Coraggio.''

> **Agent Thoughts:**
> So the next query is `<query>` What is the release year of Il Coraggio and who directed it? `</query>` Based on the query, the relevant evidence is `<evidence>`Il Coraggio is a 1955 Italian comedy film directed by Domenico Paolella. `</evidence>`

**LLM judge**

> The agent's reasoning is severely incomplete and structurally flawed. It begins with a partial query that only addresses Il Coraggio but completely ignores Shark Monroe, (Omitted for brevity)
>
> Moreover, the query does not directly request the death dates of the directors, which is the core requirement (Omitted for brevity)
>
> While the evidence retrieved is correct and related to Il Coraggio, it is insufficient (Omitted for brevity)
>
> **So the score is 15.**

Figure 8: An example of agent partial correctness in a RAG process.

## Case of LLM judge incorrect process

**Question**
``Which film has the director died later, Il Coraggio or Shark Monroe?''
**Golden Answers**
``Il Coraggio.''

> **Agent Thoughts:**
> So the next query is `<query>`When did the directors of Il Coraggio and Shark Monroe die?`</query>` Based on the query, the relevant evidence is: `<evidence>`Tonino Ricci, the director of Il Coraggio (also known as The Liberators), died on 9 March 2014. Umberto Lenzi, the director of Shark Monroe (also known as Man from the Deep River), died on 19 October 2017.`</evidence>`

**LLM judge**

> The agent's reasoning shows a series of critical errors. First, the query itself is flawed: it misattributes directors to both films. Il Coraggio (1955) was directed by Domenico Paolella, not Tonino Ricci. (Omitted for brevity)
>
> This fundamental misidentification invalidates the entire reasoning chain. The agent is not comparing the correct individuals, rendering the final answer baseless (Omitted for brevity)
>
> Given these severe and foundational flaws in both retrieval and inference,(Omitted for brevity)
>
> **So the score is 5.**

Figure 9: An example of agent incorrectness in a RAG process.

**Question**
''Which film has the director died later, Il Coraggio or Shark Monroe?''
**Golden Answers**
''Il Coraggio.''

---

**Agent Thoughts:**
So the next query is `<query>`What is the release year of Il Coraggio and who directed it?`</query>` Based on the query, the relevant evidence is `<evidence>`Il Coraggio is a 1955 Italian comedy film directed by Domenico Paolella./`<evidence>` So the next query is `<query>`Who directed Shark Monroe and when was the director's death?`</query>` Based on the query, the relevant evidence is `<evidence>`William S. Hart directed Shark Monroe, which was released in 1918. William S. Hart died on December 3, 1945.`</evidence>`So the next query is `<query>`When did Domenico Paolella die?`</query>` Based on the query, the relevant evidence is `<evidence>`Domenico Paolella died on 7 October 2002. `</evidence>`So the answer is `<answer>`Il Coraggio`</answer>`

**LLM judge**

---

The agent's reasoning process is largely correct in structure and intent: it decomposes the question into sub-queries that retrieve the directors of both films, their dates of death, and uses this information to compare who died later. This stepwise reasoning is sound and correctly arrives at the golden answer: Il Coraggio.
However, under strict scrutiny, several minor flaws emerge:

1. Imprecision in Initial Querying (Omitted for brevity)
2. Redundancy(Omitted for brevity)
3. Lack of Justification in the Final Step(Omitted for brevity)

Despite these imperfections, the agent correctly disambiguates the films and retrieves accurate information. (Omitted for brevity)

**So the score is 88.**

Figure 10: An example of agent complete correctness in a RAG process.

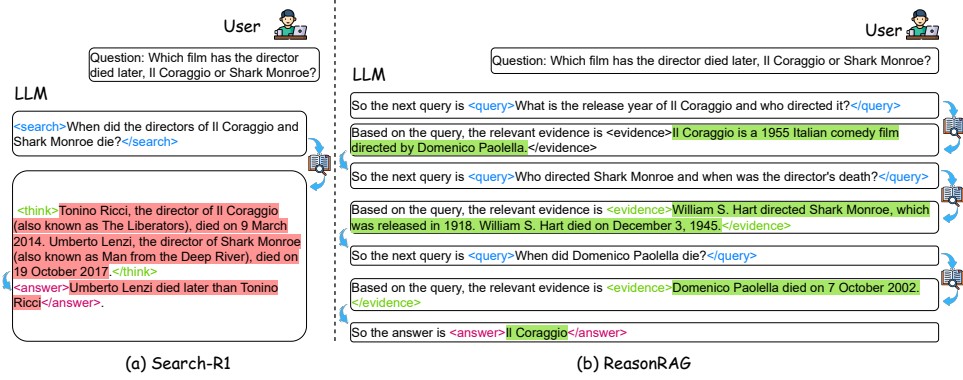

Figure 11: Case Study on 2WikiMultihopQA dataset.

# D Evaluation Dataset Details

**Evaluation Datasets.** We use the process-level annotated questions (4,603 examples) as the training set for our policy optimization. These include 704 questions from PopQA, 2,843 from HotpotQA, and 1,056 from 2WikiMultiHopQA. For evaluation, we use the remaining unlabeled samples from PopQA as the test set, and we adopt the official development splits of HotpotQA and 2WikiMultiHopQA as test sets for multi-hop reasoning evaluation. Table 4 summarizes the training and test set sizes for each source. These datasets vary in their design focus and reasoning requirements. HotpotQA and 2WikiMultiHopQA are constructed to evaluate multi-hop reasoning capabilities, where answering a question requires combining information from multiple passages. HotpotQA includes sentence-level supporting facts and diverse question types, such as bridge and comparison questions. 2WikiMulti-HopQA ensures genuine multi-step inference by leveraging structured knowledge from Wikidata and constructing explicit reasoning paths. PopQA, in contrast, is an open-domain QA dataset designed to probe factual recall in large language models. It focuses on a wide spectrum of factual knowledge, from high-frequency popular facts to long-tail, less commonly known information. By combining these datasets, we cover a diverse set of reasoning challenges, including factual retrieval, multi-hop inference, and process-level supervision.

| Dataset Source | Train Set Size | Test Set Size |
|---|---|---|
| PopQA | 704 | 11,267 |
| HotpotQA | 2,843 | 7,405 |
| 2WikiMultiHopQA | 1,056 | 12,576 |
| bamboogle | - | 125 |
| musique | - | 2,417 |
| **Total** | **4,603** | **33,790** |

Table 4: Number of examples in the training and test sets for each dataset. Process-level annotations are used for training; test sets include remaining PopQA examples and official development splits of other datasets.

**Evaluation Details.** To evaluate model performance on question answering tasks, we adopt two standard metrics: Exact Match (EM) and $F_1$ score.

**Exact Match (EM)** measures the percentage of predictions that exactly match any of the reference answers. Formally,

$$\text{EM} = \frac{1}{N} \sum_{i=1}^{N} \delta \left( y_i^{\text{pred}}, \, y_i^{\text{gold}} \right), \tag{8}$$

where $N$ is the number of examples, and $\delta(a, b) = 1$ if $a = b$, otherwise 0.

**$F_1$ score** computes the token-level overlap between the predicted answer and the ground-truth answer. Let $T_i^{\text{pred}}$ and $T_i^{\text{gold}}$ denote the sets of tokens in the predicted and gold answers, respectively:

$$\text{Precision}_i = \frac{|T_i^{\text{pred}} \cap T_i^{\text{gold}}|}{|T_i^{\text{pred}}|}, \tag{9}$$

$$\text{Recall}_i = \frac{|T_i^{\text{pred}} \cap T_i^{\text{gold}}|}{|T_i^{\text{gold}}|}, \tag{10}$$

$$\text{F}_1 = \frac{1}{N} \sum_{i=1}^{N} \frac{2 \cdot \text{Precision}_i \cdot \text{Recall}_i}{\text{Precision}_i + \text{Recall}_i}. \tag{11}$$

We follow the official evaluation metrics implementation provided by the FlashRAG toolkit [75].

# E Implementation Details

We summarize all artifacts (datasets, models, baselines, external knowledge base, etc) used in our experiments and their resource links in Table 5.

| Name | Purpose | Artifact URL |
|---|---|---|
| PopQA | Eval Dataset | `https://huggingface.co/datasets/RUC-NLPIR/FlashRAG_datasets/tree/main/popqa` |
| HotpotQA | Eval Dataset | `https://huggingface.co/datasets/RUC-NLPIR/FlashRAG_datasets/tree/main/hotpotqa` |
| 2WikiMultiHopQA | Eval Dataset | `https://huggingface.co/datasets/RUC-NLPIR/FlashRAG_datasets/tree/main/2wikimultihopqa` |
| Bamboogle | Eval Dataset | `https://huggingface.co/datasets/RUC-NLPIR/FlashRAG_datasets/tree/main/bamboogle` |
| MuSiQue | Eval Dataset | `https://huggingface.co/datasets/RUC-NLPIR/FlashRAG_datasets/tree/main/musique` |
| `RAG-ProGuide` | Train Dataset | `https://anonymous.4open.science/r/ReasonRAG-B442.` |
| BGE | Retriever | `https://huggingface.co/BAAI/bge-base-en-v1.5` |
| Wikidump 2018 | Knowledge Source | `https://archive.org/download/enwiki-20181220/enwiki-20181220-pages-articles.xml.bz2` |
| Qwen2.5-7B-Instruct | Backbone Model | `https://huggingface.co/Qwen/Qwen2.5-7B-Instruct` |
| Adaptive-RAG | Baseline | `https://huggingface.co/illuminoplanet/combined_flan_t5_xl_classifier` |
| Self-RAG | Baseline | `https://huggingface.co/selfrag/selfrag_llama2_13b` |
| AutoRAG | Baseline | `https://huggingface.co/ICTNLP/Auto-RAG-Llama-3-8B-Instruct` |
| Search-R1 | Baseline | `https://huggingface.co/PeterJinGo/SearchR1-nq_hotpotqa_train-qwen2.5-7b-it-em-ppo` |
| `ReasonRAG` | Our Method | `https://anonymous.4open.science/r/ReasonRAG-B442.` |

Table 5: Resource links of artifacts used in our experiments.

### E.1 Implementation Details of `ReasonRAG`

Following the setup in the FlashRAG toolkit, we use Wikidump 2018 as our knowledge source. To ensure retrieval quality, we augment our corpus by incorporating relevant content from the PopQA, HotpotQA, and 2WikiMultiHopQA datasets. All datasets are available on Huggingface. Subsequently, we employ BGE as our retriever, consistently retrieving the top 3 documents. For all methods not requiring fine-tuning, we use Qwen2.5-7B-Instruct as our baseline model for inference.

### E.2 Implementation Details of Baselines

For baseline implementations, we utilize the FlashRAG [75] reproduction, where several models, such as Naïve Generation, Standard RAG, FLARE, Iter-Retgen, RECOMP, LongLLMLingua, and Selective-Context, require no additional parameter configuration. For Self-RAG, we use the checkpoint provided in the FlashRAG reproduction. For AdaptiveRAG, we employ the FlashRAG reproduction's router and qwen2.5-7b-instruct as the reasoning model. For AutoRAG, we conduct inference using the publicly available checkpoint from Hugging Face. For Search-R1, we use the reproduced qwen2.5-7b-base and qwen2.5-7b-instruct checkpoints for inference.

## F Prompt Instructions

Our prompts include Reasoning, Grounding in Agentic RAG workflow, and a process evaluation prompt for node expansion. No extra prompt design is needed when input a new question into LLMs for inference. The prompt details are illustrated in Figure 12, Figure 13, and Figure 14.

## G License Agreement

The `RAG-ProGuide` is constructed based on popqa, hotpotqa, and 2wikimultihopqa from FlashRAG dataset [75]. All these datasets are using CC-BY-SA-4.0 license, allowing the modification for research use. For the new constructed dataset `RAG-ProGuide`, including but not limited to the questions and process-level reward, we make them available solely for research purposes. Users are permitted to use, modify, and share these annotations for academic and non-commercial research activities. Any other use, including commercial exploitation, is not permitted without explicit written permission from the authors.

## Reasoning

You are a question-answering assistant with access to a retrieval tool. Your goal is to provide a concise and accurate reasoning process.
Instructions:
* Error Reflection: If errors exist in previous thoughts, identify and correct them. Skip this step if no errors are present.
* Information Sufficiency: Evaluate whether the current information is sufficient to fully and accurately answer the question. If additional retrieval is needed, deconstruct the question and generate the next query. Avoid repeating previous queries. If no meaningful new query can be generated, explain why and provide an answer based on the current information. * Conciseness: Ensure both queries and answers are concise, using nouns or short phrases whenever possible.
* Conclusion:
If generating an answer: "So the answer is <answer>{answer_format}</answer>". If more retrieval is needed: "So the next query is <query>query</query>".

Figure 12: System prompt for Reasoning

## Grounding

You are an information retrieval assistant. Your task is to extract relevant evidence from the provided Wikipedia documents based on the latest query.
Instructions:
* Identify key terms or concepts in the query. * Search the documents for evidence that supports the query. * Response format: If relevant evidence is found, output: Based on the query, the relevant evidence is <evidence>evidence</evidence>. If no relevant evidence is found, output: <evidence>None</evidence>.

Figure 13: System prompt for Evidence Extraction

## Process Evaluation

An agent is tasked with answering a question using a retrieval tool. Critically assess its intermediate reasoning process to determine if it leads to the correct answer. Identify all flaws, inconsistencies, and mistakes in the thought process. Every imperfection, no matter how small, must be acknowledged. Evaluate how effectively the reasoning supports the final answer and the overall accuracy of the response. Ensure the evaluation is extremely harsh, leaving no leniency. Even if the answer seems close to correct, do not award full marks to maintain strict grading standards. Assign a score between [0, 1] based on the severity of flaws and the reasoning's accuracy in leading to the golden answer. Respond briefly and conclude with: So the score is [Score].

Figure 14: System prompt for Process Evaluation

# H Limitations

We employ process-supervised RL to optimize the model policy. In contrast to outcome-supervised RL, our approach necessitates exploring process-level actions for fine-grained reward annotation. Consequently, acquiring sufficient data for process-level annotation incurs a higher time cost compared to outcome supervision during data rollout. Nevertheless, as the training efficiency verifies, our data exhibits superior quality, enabling models to achieve greater performance gains with fewer data samples.

# I Societal Impacts

LLMs carry the risk of generating uncontrollable responses. When an LLM retrieves racist or harmful information from a search engine, it could be inadvertently led to produce similar content. We strongly advise users to employ our agentic RAG framework responsibly by integrating secure search engines or knowledge corpora and conducting evaluations within open-source, safe environments and datasets.

