# OpenReview forum: "Process vs. Outcome Reward: Which is Better for Agentic RAG Reinforcement Learning"
_NeurIPS.cc/2025/Conference — NeurIPS 2025 poster_

### Official Review · Reviewer_aFno · 2025-07-01

**Clarity:** 3
**Significance:** 3
**Originality:** 3
**Rating:** 4
**Confidence:** 4

**Summary:**

The paper investigates reward design for agentic RAG. It introduces ReasonRAG, which uses Monte-Carlo Tree Search (MCTS) plus a “Shortest-Path Reward Estimation” (SPRE) signal to build a 5k-query process-level dataset (RAG-ProGuide). A policy trained with DPO on these preference pairs outperforms the outcome-level baseline Search-R1 on five QA benchmarks while using less data.

**Questions:**

1. Effect of data selection on sample efficiency: The method achieves strong results using only 5k queries, compared to 90k used in Search-R1. This raises the question: to what extent does the performance stem from careful data selection rather than reward formulation? How were these 5k examples chosen, and could biased or curated selection influence the outcome?

2. Does SPRE provide sufficiently discriminative reward signals for complex queries where multiple reasoning paths may be valid? Are there failure cases where SPRE assigns similar scores to semantically different strategies?

3. How reliable are the preference labels used for DPO? Were the "better" and "worse" trajectories chosen purely via SPRE scores?

**Ethical Concerns:**

["NO or VERY MINOR ethics concerns only"]

**Final Justification:**

This paper proposed reward design for agentic RAG with MCTS. Even though some of the experiments in the first draft are not solid, after discussion, the ablation studies in the author's responses have shown the effectiveness of the reward and training. I think I will keep my positive score.
Some reviewers were concerned about the MCTS latency and novelty and the author answered these questions but seems not convinced the other author.

**Limitations:**

See weakness.

**Paper Formatting Concerns:**

No.

**Quality:**

3

**Strengths And Weaknesses:**

Strength:
1. The paper addresses an important problem in agentic retrieval-augmented generation—how to provide effective reward beyond sparse final answer.
2. The proposed model outperforms a 90k-sample outcome-based baseline across five QA benchmarks with only 5k sample.

Weakness:
1. Component-wise contribution unclear: The proposed system integrates several components—MCTS, SPRE, and DPO—but their relative importance remains unclear. An ablation study removing or replacing each module (e.g., substituting MCTS with naive sampling) would help clarify which elements are essential to performance.

2. Penalty coefficient \alpha is under-analyzed: Equation 1 includes a tunable penalty weight \alpha to encourage shorter paths. However, the paper does not analyze its effect on learning dynamics or final performance. Could the authors provide experiments or sensitivity analysis on different values of \alpha?

3. Process vs. outcome comparison not controlled: Search-R1 differs in model scale and data quantity. To isolate reward type effects, would it be possible to train a DPO model on outcome-based preferences from the same 5k questions?

---

> ### Author Rebuttal · Authors · 2025-07-31
>
> Dear Reviewer aFno,
>
> We sincerely thank you for your thoughtful review and valuable feedback on our work. Below, we address each of your concerns with additional experimental evidence and clarifications.
>
> ## **1. Ablation Study (MCTS, SPRE, DPO)**
> > Component-wise contribution unclear: The proposed system integrates several components—MCTS, SPRE, and DPO—but their relative importance remains unclear. An ablation study removing or replacing each module (e.g., substituting MCTS with naive sampling) would help clarify which elements are essential to performance.
>
> > Process vs. outcome comparison not controlled: Search-R1 differs in model scale and data quantity. To isolate reward type effects, would it be possible to train a DPO model on outcome-based preferences from the same 5k questions?
>
> Thank you for the valuable suggestion. As mentioned in Table 3, we conducted a comprehensive ablation study to evaluate the contributions of MCTS, reward design, and training methods, as detailed below:
>
> ### **1.1 Ablation study settings**
>
> | Variant                        | MCTS | Reward         | Training |
> | ------------------------------ | ---- | -------------- | -------- |
> | ReasonRAG (Base)               | ✗    | ✗              | ✗       |
> | ReasonRAG (SFT)                | ✓    | SPRE           | SFT      |
> | ReasonRAG (RL-ORL)             | ✗    | Outcome reward | GRPO     |
> | ReasonRAG (RL-PRL)             | ✓    | SPRE           | DPO      |
> | ReasonRAG (Outcome reward DPO) | ✓    | Outcome reward | DPO      |
> | ReasonRAG (RL-PRL) | ✓    | Outcome reward | DPO      |
>
> Following your valuable advice, we added a controlled experiment replacing our process-level reward with a simplified final-answer-based reward (Outcome reward DPO). The results are as follows:
>
> ### **1.2 Performance Comparison**
> | Method                          | PopQA EM | PopQA F1 | HotpotQA EM | HotpotQA F1 | 2WikiMulti EM | 2WikiMulti F1 | Bamboogle EM | Bamboogle F1 | MuSiQue EM | MuSiQue F1 | Avg. EM  | Avg. F1  |
> | ------------------------------- | -------- | -------- | ----------- | ----------- | ------------- | ------------- | ------------ | ------------ | ---------- | ---------- | -------- | -------- |
> | ReasonRAG (Base)                | 35.6     | 42.7     | 23.7        | 38.2        | 15.2          | 28.9          | 28.0         | 38.7         | 7.7        | 15.4       | 22.0     | 32.8     |
> | ReasonRAG (SFT)                 | 31.6     | 37.4     | 26.8        | 38.7        | 35.1          | 40.9          | 17.6         | 27.3         | 8.6        | 15.5       | 23.9     | 32.0     |
> | ReasonRAG (RL-ORL) (5k queries) | 23.0     | 30.9     | 28.1        | 32.6        | 32.0          | 43.8          | 17.5         | 24.1         | 5.9        | 13.1       | 21.3     | 28.9     |
> | ReasonRAG (Outcome reward DPO)  | 33.0     | 43.4     | 29.4        | 41.3        | 35.8          | 41.1          | 28.0         | 39.3         | 12.5       | 21.8       | 27.7     | 37.4     |
> | ReasonRAG (RL-PRL)              | **41.5** | **46.2** | **38.4**    | **48.9**    | **43.6**      | **50.4**      | **36.0**     | **45.5**     | **12.8**   | **20.6**   | **34.5** | **42.3** |
>
> ### **1.3 Key findings**
> (1) SPRE enhances performance compared to outcome-only reward designs.
> (2) MCTS-guided data generation contributes to stronger reward signals and sample efficiency, which boosts performance especially when combined with process-level reward supervision.
> (3) Base and SFT variants show limited gains, highlighting the importance of both reward quality and training method.
>
> ## **2. Penalty Analysis**
>
> > Penalty coefficient \alpha is under-analyzed: Equation 1 includes a tunable penalty weight \alpha to encourage shorter paths. However, the paper does not analyze its effect on learning dynamics or final performance. Could the authors provide experiments or sensitivity analysis on different values of \alpha?
>
> Thank you for your comment. Our reward design follows two key principles: (1) reasoning chains with higher accuracy should receive higher rewards, and (2) among chains with similar accuracy, shorter reasoning paths should be rewarded more. To achieve this, we set the penalty coefficient α within the range (0,1], choosing a value close to 1 to effectively balance these objectives.
>
> ## **3. Sample Efficiency**
> > Effect of data selection on sample efficiency: The method achieves strong results using only 5k queries, compared to 90k used in Search-R1. This raises the question: to what extent does the performance stem from careful data selection rather than reward formulation? How were these 5k examples chosen, and could biased or curated selection influence the outcome?
>
> Thank you for the thoughtful question. As mentioned on Page 5, Lines 165–167, we randomly sampled 3k examples from each of the three datasets, applied MCTS-based filtering, and retained 5k high-quality trajectories. Therefore, the performance gains are not due to curated data, but rather the effectiveness of our automated data generation pipeline. This pipeline adaptively selects diverse, high-reward trajectories through search and reward modeling, enabling strong performance with limited supervision.
>
> ## **4. Reward reliability**
> > Does SPRE provide sufficiently discriminative reward signals for complex queries where multiple reasoning paths may be valid? Are there failure cases where SPRE assigns similar scores to semantically different strategies?
>
> Thank you for your comment. SPRE provides sufficiently informative reward signals for training. While semantically different reasoning paths may receive similar SPRE scores, we address this by filtering preference pairs based on their reward gap, as described in Figure 5 and Lines 172–175:
> *“To ensure high-quality preference data, we perform post-processing to remove duplicates and uninformative comparisons: (i) we discard preference pairs with identical response sequences, and (ii) pairs with a reward difference less than 0.01. After filtering, the final dataset consists of 4,603 questions and 13,289 distinct preference pairs."*.
>
> This ensures that retained comparisons exhibit clear differences, enhancing the discriminative strength of the supervision.
>
> ## **5. Label Annotation**
> > How reliable are the preference labels used for DPO? Were the "better" and "worse" trajectories chosen purely via SPRE scores?
>
> ### **5.1 SPRE Annotation**
> Thank you for the comment. Preference labels are assigned **purely based on SPRE scores**, which prioritize both correctness and shorter reasoning paths. To ensure reliability, we filter out preference pairs with reward gaps below 0.01, and report the distribution of reward gaps in Figure 3(d).
> ### **5.2 Open-Source Dataset and Manual Inspection**
> **The full dataset is available via a link in Appendix Table 5**, and manual inspection confirms that SPRE-based rewards reliably reflect reasoning quality.
>
> We hope these responses address your concerns comprehensively. Feel free to let us know if you have any additional questions, and we are more than happy to provide further clarification on any aspect of our work.
>
> Best regards,
>
> The Authors

---

> > ### Comment · Reviewer_aFno · 2025-08-01
> > **Concern solved**
> >
> > Thank you for the detailed and thoughtful rebuttal. I appreciate the comprehensive ablation study and the added controlled comparison between process- and outcome-level reward training, which address the main points raised in the initial review.
> >
> > That said, I would like to point out two remaining minor issues:
> >
> > - Variant naming ambiguity: In Table 3 (1.1), the label “ReasonRAG (RL-PRL)” appears twice, both seemingly using outcome rewards and DPO. This could be a typographical error, but clarification would improve reproducibility.
> >
> > - Lack of empirical analysis of the penalty coefficient $\alpha$: While you explained the design motivation behind $\alpha$, there is no accompanying sensitivity analysis to show how this parameter affects learning dynamics or final performance. Given its central role in the reward design, such analysis would strengthen the empirical grounding of your approach.
> >
> > These are relatively minor concerns, and overall your rebuttal improves the clarity and completeness of the submission. Thank you again for your careful response.

---

> > > ### Author Response · Authors · 2025-08-01
> > > **Further Clarification**
> > >
> > > Dear Reviewer aFno,
> > >
> > > We sincerely thank you for your careful review and thoughtful comments. We greatly appreciate the time and effort you have dedicated to helping us improve our work.
> > >
> > > ## **1. ReasonRAG(RL-PRL) Setting**
> > > > Variant naming ambiguity: In Table 3 (1.1), the label “ReasonRAG (RL-PRL)” appears twice, both seemingly using outcome rewards and DPO. This could be a typographical error, but clarification would improve reproducibility.
> > >
> > > We sincerely apologize for the mistake in reporting ReasonRAG (RL-PRL) twice in the original table. To clarify, the correct setting for ReasonRAG (RL-PRL) is as follows:
> > >
> > > | Variant            | MCTS | Reward | Training |
> > > | ------------------ | ---- | ------ | -------- |
> > > | ReasonRAG (RL-PRL) | ✓    | SPRE   | DPO      |
> > >
> > > We appreciate your understanding and thank you for helping us improve the clarity and reproducibility of our work.
> > >
> > > ## **2. Penaly Analysis**
> > > > Lack of empirical analysis of the penalty coefficient $\alpha$: While you explained the design motivation behind $\alpha$, there is no accompanying sensitivity analysis to show how this parameter affects learning dynamics or final performance. Given its central role in the reward design, such analysis would strengthen the empirical grounding of your approach.
> > >
> > > Thank you for your valuable feedback. We have included the relevant sensitivity analysis in our latest update to better illustrate the impact of $\alpha$ on performance.
> > >
> > > We hope these responses address your concerns comprehensively. Feel free to let us know if you have any additional questions, and we are more than happy to provide further clarification on any aspect of our work.
> > >
> > > Best regards,
> > >
> > > The Authors

---

> > > ### Author Response · Authors · 2025-08-05
> > > **Clarifications on Remaining Concerns**
> > >
> > > Dear Reviewer aFno,
> > >
> > > Thank you very much for your thoughtful follow-up and for pointing out these remaining concerns.
> > >
> > > We have addressed both issues in our latest response. Specifically, we clarified the variant naming ambiguity in Table 3 (1.1). Additionally, we included a sensitivity analysis of the penalty coefficient to better illustrate its effect on learning dynamics and final performance, as suggested.
> > >
> > > If there are any aspects that remain unclear or would benefit from further clarification, we would be more than happy to elaborate. We greatly appreciate your time, careful reading, and constructive suggestions throughout this process.

---

> > > ### Author Response · Authors · 2025-08-07
> > > **Re: Concern solved**
> > >
> > > Dear Reviewer aFno,
> > >
> > > Thank you very much for your thoughtful and constructive feedback. We are glad to hear that the ablation study and the added comparisons addressed the main concerns in your initial review.
> > >
> > > Regarding the two minor issues you kindly pointed out:
> > >
> > > - Variant naming ambiguity: We have clarified this in the latest version of our rebuttal.
> > >
> > > - Penalty coefficient analysis: We appreciate the suggestion. In our updated rebuttal, we have added a brief empirical sensitivity analysis to illustrate how this parameter influences model performance, thus supporting the design choice more concretely.
> > >
> > > As the discussion deadline is approaching, we would like to kindly ask whether there are any remaining concerns that we may not have addressed fully. We are more than happy to provide further clarification if needed.
> > >
> > > If you feel that the concerns have been adequately addressed, we would sincerely appreciate it if you could consider a score revision. Your feedback is very important to us and greatly valued.
> > >
> > > Thank you again for your time and thoughtful engagement.
> > >
> > > Best regards,
> > >
> > > Authors

---

> > > > ### Comment · Reviewer_aFno · 2025-08-07
> > > > **Response for final decision**
> > > >
> > > > I have read all your updated responses, and most of the concerns are clarified. So I will reclaim my positive score.

---

> > > > > ### Author Response · Authors · 2025-08-07
> > > > > **Appreciation for Your Thoughtful Reassessment**
> > > > >
> > > > > Dear Reviewer aFno,
> > > > >
> > > > > Thank you for your positive feedback and for updating us on the score. We genuinely appreciate your insightful and constructive comments. We truly appreciate your thoughtful reconsideration and are grateful for the multiple rounds of communication, as your engagement has greatly enhanced the quality of our work.
> > > > >
> > > > > Best,
> > > > >
> > > > > Authors

---

> > ### Author Response · Authors · 2025-08-01
> > **Further Clarifications and Experimental Support**
> >
> > Dear Reviewer aFno,
> >
> > We sincerely thank you for your thoughtful review and valuable feedback on our work. Due to time constraints, our initial response did not fully address all experimental aspects. Below, we address each of your concerns with additional experimental evidence and clarifications.
> >
> > ## **1. Penalty Analysis**
> >
> > > Penalty coefficient \alpha is under-analyzed: Equation 1 includes a tunable penalty weight \alpha to encourage shorter paths. However, the paper does not analyze its effect on learning dynamics or final performance. Could the authors provide experiments or sensitivity analysis on different values of \alpha?
> >
> > Thank you for your valuable comment. To study the effect of \alpha, we conducted an additional experiment using \alpha=0.5, while keeping all other training configurations fixed. The results are shown below:
> >
> > | Setting | PopQA EM | PopQA F1 | HotpotQA EM | HotpotQA F1 | 2Wiki EM | 2Wiki F1 | Bamboogle EM | Bamboogle F1 | MuSiQue EM | MuSiQue F1 | Avg. EM| Avg. F1|
> > | ------------------------- | -------- | -------- | ----------- | ----------- | -------- | -------- | ------------ | ------------ | ---------- | ---------- | -------- | -------- |
> > | $\alpha = 0.5$| 36.8 | 41.9 | 30.2| 39.7| 33.8 | 40.9 | 27.1 | 35.6 | 8.6| 14.3 | 27.3 | 34.5 |
> > | **$\alpha = 0.9$ (ours)** | **41.5** | **46.2** | **38.4**| **48.9**| **43.6** | **50.4** | **36.0** | **45.5** | **12.8** | **20.6** | **34.5** | **42.3** |
> >
> > ### **1.1 Key findings**
> > We observe a significant performance drop in both EM and F1 across datasets when reducing α to 0.5. This supports our claim that a higher penalty (e.g., \alpha=0.9) is crucial for balancing correctness and path efficiency, especially in complex reasoning settings.
> >
> > ## **2. Sample Efficiency**
> > > Effect of data selection on sample efficiency: The method achieves strong results using only 5k queries, compared to 90k used in Search-R1. This raises the question: to what extent does the performance stem from careful data selection rather than reward formulation? How were these 5k examples chosen, and could biased or curated selection influence the outcome?
> >
> > Thank you for the thoughtful question. Our goal was not to handpick high-performing queries, but to build a **diverse, high-signal training set** in a scalable and reproducible manner.
> >
> > As described on Page 5, Lines 165–167, we first **randomly sampled 3,000 questions from each of the three datasets** (PopQA, HotpotQA, and 2WikiMultiHopQA) to ensure dataset coverage. We then applied **MCTS-based trajectory exploration** on these questions to generate a diverse pool of reasoning paths per query. Among these, we filtered the reasoning data based on SPRE reward gap and retained the **top 5,000 quries**.
> >
> > This process is fully automated and aims to provide high-quality supervision without human bias. In contrast to manually curated data, which is often costly, time-consuming, and difficult to scale, our approach offers a scalable and efficient alternative. As a result, the performance gains mainly stem from the reward design and search-guided sampling rather than from careful manual curation.
> >
> > ## **3. Reward reliability**
> > > Does SPRE provide sufficiently discriminative reward signals for complex queries where multiple reasoning paths may be valid? Are there failure cases where SPRE assigns similar scores to semantically different strategies?
> >
> > > How reliable are the preference labels used for DPO? Were the "better" and "worse" trajectories chosen purely via SPRE scores?
> >
> > Thank you for your valuable feedback. To further validate the reliability of SPRE-based preference labels, we conducted a manual evaluation on 200 randomly sampled preference pairs. Five PhD-level annotators independently assessed whether the SPRE-chosen trajectory was truly better than the rejected one. The evaluation focused on the following four dimensions:
> >
> > | Metric| Description | Agreement Rate |
> > | --------------------------------- | ------------------------------------------------------------------------------------------- | -------------- |
> > | **Correctness Preference**| Whether the chosen response is more factually accurate and complete | 98.5%|
> > | **Conciseness Preference**| Whether the reasoning is clearer and more succinct| 94.0%|
> > | **Faithfulness Preference** | Whether the chosen response aligns better with retrieved evidence | 96.0%|
> > | **Overall Preference Confidence** | At least 4 out of 5 annotators prefer chosen over rejected | 95.5%|
> >
> > ### **3.1 Key findings**
> > These results confirm that SPRE-based preferences are highly consistent with human judgment, providing reliable supervision for DPO training.
> >
> > We hope these responses address your concerns comprehensively. Feel free to let us know if you have any additional questions, and we are more than happy to provide further clarification on any aspect of our work.
> >
> > Best regards,
> >
> > The Authors

---

### Official Review · Reviewer_2RBG · 2025-07-03

**Clarity:** 3
**Significance:** 3
**Originality:** 1
**Rating:** 4
**Confidence:** 4

**Summary:**

The paper examines the impact of ORM and PRM on RAG FT by creating an MCTS‑sampled dataset, RAG‑ProGuide, and training the inference model, ReasonRAG, via DPO FT. The proposed workflow demonstrates innovation in exploration diversity, iteration‑count distribution, and performance on downstream datasets. Furthermore, the authors compare ReasonRAG’s downstream metrics after ORM versus PRM FT across multiple datasets, providing preliminary evidence that PRM outperforms ORM and offering valuable insights for RAG researche

**Questions:**

1. Given that the primary focus of your paper is the comparative impact of PRM versus ORM on LLM FT, it is advisable to enrich the PRM experiments, for example, by assigning distinct weights to procedural correctness and outcome correctness and systematically evaluating how these weightings affect downstream performance metrics.

2. The set of baselines should concentrate on RAG frameworks that have undergone comparable ORM or PRM FT. Incorporating additional relevant baselines, such as RAG‑Gym, would further bolster the robustness and persuasiveness of your evaluation.

3. The axis labels and their semantic descriptions in certain figures (e.g., Figure 3) lack sufficient clarity. A careful revision of these annotations is necessary to ensure that the figure’s intent and interpretation are unambiguous.

**Ethical Concerns:**

["NO or VERY MINOR ethics concerns only"]

**Final Justification:**

The authors have provided the following responses or adjustments:
1.Clarified the distinctions from other similar works;
2.Explained the content of the figures and committed to making corrections in the future.
3.Added experiments with a new model (Qwen2.5-3B) and algorithm (Search-O1).
They address some of my concerns, and I decided to slightly increase the score.

**Limitations:**

yes

**Quality:**

2

**Strengths And Weaknesses:**

> Strengths:

1.The paper’s premise—comparing the impact of PRM versus ORM on agentic RAG FT is both novel and of practical significance to the RAG research community.

2.The experimental evaluation is thorough, leveraging a diverse set of datasets and RAG frameworks to assess exploration diversity, iteration‑count distributions, and test‑set EM performance; these comprehensive results effectively demonstrate the advantages of ReasonRAG.

> Weaknesses:

1.From a workflow perspective, an MCTS‑based RAG system has already been proposed in AirRAG (https://arxiv.org/pdf/2501.10053). The main contribution of this paper—using a UCB‑based stepwise selection of positive and negative examples for DPO FT offers limited novelty.

2.In terms of training loss, although the paper’s focus is on comparing PRM and ORM for agentic RAG FT, it applies identical DPO weighting to both procedural decision correctness and final‑output correctness. The manuscript does not explore how varying these weightings affects training outcomes, nor does it investigate which aspect has a greater impact on RAG system performance.

3.Figure 3 contains incorrect annotations, and its caption does not clearly explain the figure’s intended meaning.

4.The paper does not specify which large language models were used at each baseline; if different LLMs were employed, the reliability of the results would be compromised.

---

> ### Author Rebuttal · Authors · 2025-07-31
>
> Dear Reviewer 2RBG,
>
> We sincerely thank you for your thoughtful review and valuable feedback on our work. Below, we address each of your concerns with additional experimental evidence and clarifications.
>
> ## **1. ReasonRAG vs. AirRAG**
> > From a workflow perspective, an MCTS‑based RAG system has already been proposed in AirRAG (https://arxiv.org/pdf/2501.10053). The main contribution of this paper—using a UCB‑based stepwise selection of positive and negative examples for DPO FT offers limited novelty.
>
> Thank you for the valuable comment. We agree that MCTS has previously been explored in RAG contexts. Our approach differs fundamentally from AirRAG in goals, methodology, and the role of MCTS. A comparison is summarized below:
>
> | Aspect   | **ReasonRAG (Ours)**     | **AirRAG**      |
> | ---------------------- | -------------------------------------------------------- | --------------------------------------------------------------- |
> | **Training Paradigm** | PRL (DPO with process-level rewards) | SFT     |
> | **Inference** | Agentic RAG (multi-step planning and retrieval)  | MCTS + Self-consistency decoding    |
> | **Role of MCTS** | Offline data generation for training high-quality traces | Online inference-time search and answer selection  |
> | **Research Focus** | Improving **training efficiency** and agentic reasoning | Enhancing **inference-time** reasoning accuracy |
>
> While both systems utilize MCTS, **ReasonRAG leverages it during training to enable reward-driven supervision**, whereas **AirRAG applies MCTS at inference to aggregate final answers via self-consistency**. We sincerely appreciate the reviewer for highlighting this connection, as it helped us better position our contributions relative to existing work.
>
> ## **2. Step-Level vs. Outcome-Level Supervision**
> > In terms of training loss, although the paper’s focus is on comparing PRM and ORM for agentic RAG FT, it applies identical DPO weighting to both procedural decision correctness and final‑output correctness. The manuscript does not explore how varying these weightings affects training outcomes, nor does it investigate which aspect has a greater impact on RAG system performance.
>
> > Given that the primary focus of your paper is the comparative impact of PRM versus ORM on LLM FT, it is advisable to enrich the PRM experiments, for example, by assigning distinct weights to procedural correctness and outcome correctness and systematically evaluating how these weightings affect downstream performance metrics.
>
> Thank you for raising this important question. We agree that identifying the relative contribution of step-level (procedural) and answer-generation(outcome) supervision is critical to understanding what drives performance in agentic RAG systems.
>
> Following your valuable suggestion, we conducted a controlled ablation by reducing the amount of training data used for either (1) step-level rewards or (2) answer-generation rewards to 50%. The variant settings are: (1) 50% Step-level Data: 4.5 step-level data + 4k answer-generation data (2) 50% Answer-Generation Data: 9k step-level data + 2k answer-generation data. The results are presented below:
>
> | Variant   | PopQA EM | PopQA F1 | HotpotQA EM | HotpotQA F1 | 2Wiki EM | 2Wiki F1 |
> | --------------------------- | -------- | -------- | ----------- | ----------- | -------- | -------- |
> | 50% Step-level Data | 35.2 | 40.3 | 29.4 | 42.0 | 30.0 | 43.1 |
> | 50% Answer-Generation Data | 38.9 | 44.2 | 35.7 | 47.0 | 39.8 | 48.5 |
> | **Full Data (Original)** | **41.5** | **46.2** | **38.4** | **48.9** | **43.6** | **50.4** |
>
> ### **2.1 Key findings**
>
> The experiment results indicates that **reducing step-level data** causes a more substantial drop. In contrast, cutting final-answer data leads to milder degradation. These results demonstrate that step-level supervision plays a more critical role in agentic RAG training.
>
> ## **3. Clarification of Figure 3**
>
> > Figure 3 contains incorrect annotations, and its caption does not clearly explain the figure’s intended meaning.
>
> > The axis labels and their semantic descriptions in certain figures (e.g., Figure 3) lack sufficient clarity. A careful revision of these annotations is necessary to ensure that the figure’s intent and interpretation are unambiguous.
>
> **Response**: Thank you for pointing this out. We apologize for the lack of clarity in the original caption and presentation. While we provided a description in the original manuscript (Page 5, Lines 180–189), we realize it may have been insufficient. We clarify the intention of Figure 3 below:
>
> ### **3.1 Pair Type Distribution**
> **Figure 3(a)** shows the distribution of preference pair types. The x-axis denotes combinations of accepted vs. rejected action types (A: answer generation, Q: query generation, E: evidence extraction), and the y-axis indicates the number of preference samples per type. This demonstrates that our dataset spans diverse comparison scenarios across multiple reasoning stages.
>
> ### **3.2 Reasoning Step Distribution**
> **Figure 3(b)** illustrates the distribution of reasoning iteration counts per query. The x-axis is the number of MCTS steps, and the y-axis is the number of samples. This reflects the varying difficulty of different multi-hop questions.
>
> ### **3.3 Token Counts Distribution**
> **Figure 3(c)** shows the distribution of response token lengths, where the x-axis is the number of tokens in generated responses and the y-axis is the sample count. This confirms our dataset captures both concise and lengthy trajectories.
>
> ### **3.4 Reward Gap Distribution**
> **Figure 3(d)** presents the distribution of reward gaps between preferred and rejected paths. The x-axis indicates the reward difference, and the y-axis shows probability density. Larger gaps indicate clearer supervision signals for preference learning.
>
> We will correct them in the revised version. Thank you again for your careful reading and helpful suggestion.
>
> ## **4. Training on more LLM variant: Qwen2.5-3B**
> > The paper does not specify which large language models were used at each baseline; if different LLMs were employed, the reliability of the results would be compromised.
>
> Thank you for the valuable comment. As mentioned in **Page 6, Lines 241–243**, all baselines in our main experiments use **Qwen2.5-7B-Instruct** as the backbone model to ensure a fair comparison. Implementation details for each baseline are provided in **Appendix F.2**, and the necessary artifacts are listed in **Appendix Table 5 (Page 20)** for full reproducibility.
> To further assess the scalability of ReasonRAG, we additionally conducted experiments using a smaller model (**Qwen2.5-3B-Instruct**). Results are shown below:
>
> | Method  | PopQA EM | PopQA F1 | HotpotQA EM | HotpotQA F1 | 2WikiMulti EM | 2WikiMulti F1 | Bamboogle EM | Bamboogle F1 | MuSiQue EM | MuSiQue F1 | Avg. EM | Avg. F1 |
> |---------------------|----------|----------|--------------|--------------|----------------|----------------|----------------|----------------|--------------|--------------|---------|---------|
> | Naïve Generation | 10.8 | 12.6 | 14.9  | 16.7  | 10.0  | 15.3  | 2.4  | 14.9  | 2.0  | 4.6  | 8.0 | 12.8 |
> | Standard RAG | 32.4 | 36.6 | 25.6  | 33.6  | 22.0  | 27.5  | 4.0  | 12.0  | 3.5  | 9.3  | 17.5 | 23.8 |
> | Search-O1  | 21.6 | 25.9 | 15.1  | 25.0  | 14.7  | 22.7  | 6.4  | 14.0  | 2.5  | 8.0  | 12.1 | 19.1 |
> | Search-R1 (90k) | 30.0 | 35.0 | 30.4  | 36.0  | 26.0  | 31.8  | 12.5  | 19.0  | 7.6  | 13.0  | 21.3 | 27.0 |
> | **ReasonRAG (5k)** | **32.9** | **36.8** | **30.0** | **41.3** | **26.6** | **32.1** | **13.6*** | **23.7*** | _6.9_ | **13.2** | **22.0** | **29.42** |
>
> ### **4.1 Key findings**
> ReasonRAG consistently outperforms baselines even with smaller LLMs, demonstrating that our method generalizes well across model sizes.
>
> ## **5. Results of Additional Baseline: Search-O1**
>
> > The set of baselines should concentrate on RAG frameworks that have undergone comparable ORM or PRM FT. Incorporating additional relevant baselines, such as RAG‑Gym, would further bolster the robustness and persuasiveness of your evaluation.
>
> **Response**: Thank you for your suggestion. Following your advice, we have included the Search-O1 baseline from RAG-Gym, and the results are presented below:
> | Method  | PopQA EM | PopQA F1 | HotpotQA EM | HotpotQA F1 | 2Wiki EM | 2Wiki F1 | Bamboogle EM | Bamboogle F1 | Musique EM | Musique F1 | Avg. EM | Avg. F1 |
> |--------------------|----------|----------|-------------|-------------|----------|----------|---------------|----------------|------------|-------------|---------|---------|
> | Search-O1  | 33.2 | 40.3 | 24.8 | 38.1 | 16.4 | 27.1 | 30.4  | 40.6  | 6.3 | 13.7 | 22.2 | 31.96 |
> | Search-R1 (90k) | 39.7 | 44.8 | 37.0 | 47.0 | 41.4 | 48.0 | 32.0  | 43.8  | **14.6** | 19.9 | 32.8 | 40.7 |
> | **ReasonRAG (5k)** | **41.5** | **46.2** | **38.4** | **48.9** | **43.6** | **50.4** | **36.0** | **45.5** | 12.8 | **20.6** | **34.4**| **42.3**|
>
> ### **5.1 Key findings**
> The experimental results demonstrate that **ReasonRAG consistently outperforms both Search-R1 and Search-O1**, further confirming the effectiveness of our approach.
>
> We hope these responses address your concerns comprehensively. Feel free to let us know if you have any additional questions, and we are more than happy to provide further clarification on any aspect of our work.
>
> Best regards,
>
> The Authors

---

> > ### Comment · Reviewer_2RBG · 2025-08-05
> >
> > Thank you for the detailed experimental additions and discussion, which have addressed some of my concerns. I have accordingly adjusted the score.

---

> > > ### Author Response · Authors · 2025-08-05
> > > **Grateful for Feedback and Open to Further Clarifications**
> > >
> > > Dear Reviewer 2RBG,
> > >
> > > Thank you very much for your time and for adjusting the score. We sincerely appreciate your thoughtful consideration and the opportunity to clarify our contributions.
> > >
> > > If there are any remaining concerns that we may not have fully addressed, we would be truly grateful for the chance to provide further clarification. Please do not hesitate to let us know if there is anything we can elaborate on.
> > >
> > > Thank you again for your valuable feedback.

---

### Official Review · Reviewer_WaUS · 2025-07-03

**Clarity:** 3
**Significance:** 2
**Originality:** 3
**Rating:** 3
**Confidence:** 3

**Summary:**

This paper presents an algorithm for computing process-supervised rewards in agentic retrieval-augmented generation (RAG) tasks. The proposed reward formulation is integrated with reinforcement learning to train a RAG agent. Experiments on several public benchmarks demonstrate the effectiveness of the approach.

**Questions:**

1. What are the computational costs of the different algorithms listed in Tables 2 and 3? A comparison in terms of runtime or sample efficiency would help contextualize the performance gains.

2. How do computational cost and accuracy vary with the number of steps in a sample? Since the current datasets contain at most five steps, it would be valuable to understand how the method scales with longer reasoning trajectories.

3. Have you tried using different base models? It would be helpful to know whether the observed improvements are consistent across models of varying sizes or architectures.

**Ethical Concerns:**

["NO or VERY MINOR ethics concerns only"]

**Limitations:**

While the results are strong, the use of MCTS introduces additional sampling overhead. It remains unclear whether the performance gains stem primarily from the increased number of generated samples or from the specific reward computation method itself. A more controlled analysis would help clarify this distinction.

Moreover, the datasets used in their experiments involve no more than five steps per episode. It remains unclear how the computational cost and accuracy would scale as the number of steps increases.

**Quality:**

2

**Strengths And Weaknesses:**

Strengths:
The paper is clearly written, and the idea of using Monte Carlo Tree Search (MCTS) to compute process-supervised rewards is interesting. The experimental results are promising and suggest the approach is effective.

Weaknesses:
While the results are strong, the use of MCTS introduces additional sampling overhead. It remains unclear whether the performance gains stem primarily from the increased number of generated samples or from the specific reward computation method itself. A more controlled analysis would help clarify this distinction.

Moreover, the datasets used in their experiments involve no more than five steps per episode. It remains unclear how the computational cost and accuracy would scale as the number of steps increases.

---

> ### Author Rebuttal · Authors · 2025-07-31
>
> Dear Reviewer WaUS,
>
> We sincerely thank you for your thoughtful review and valuable feedback on our work. Below, we address each of your concerns with additional experimental evidence and clarifications.
>
> ## **1. Gains from Sampling Number or Reward Design?**
>
> > While the results are strong, the use of MCTS introduces additional sampling overhead. It remains unclear whether the performance gains stem primarily from the increased number of generated samples or from the specific reward computation method itself. A more controlled analysis would help clarify this distinction.
>
> Thank you for your valuable comment. We conducted two controlled experiments to disentangle **the effects of MCTS iteration count** and **reward design**. The results are summarized below:
>
> | Method   | PopQA EM | PopQA F1 | HotpotQA EM | HotpotQA F1 | 2Wiki EM | 2Wiki F1 | Bamboogle EM | Bamboogle F1 | Musique EM | Musique F1 | Avg. EM  | Avg. F1  |
> | -------- | -------- | -------- | ----------- | ----------- | -------- | -------- | ------------ | ------------ | ---------- | ---------- | -------- | -------- |
> | 32 iters(PRL) | 35.7     | 40.3     | 31.8        | 42.3        | 39.6     | 45.5     | 26.4         | 35.4         | 10.3       | 17.1       | 28.8     | 36.1     |
> | 64 iters(ORL)      | 33.0     | 43.4     | 29.4        | 41.3        | 35.8     | 41.1     | 28.0         | 39.3         | 12.5       | 21.8       | 27.7     | 37.4     |
> | 64 iters(PRL, Ours) | **41.5** | **46.2** | **38.4**    | **48.9**    | **43.6** | **50.4** | **36.0**     | **45.5**     | **12.8**   | **20.6**   | **34.5** | **42.3** |
>
> ### **1.1 Key findings**
> (1) Reducing MCTS iterations from 64 to 32 per query results in a clear performance drop, indicating that **sufficient sampling iteration improve data quality**. As also observed in Luo et al. (2024), increasing the number of MCTS samples generally improves reasoning quality, but the marginal gain diminishes beyond a certain point. We adopt 64 iterations as a practical trade-off between performance and efficiency.
>
> (2) We replaced our step-level MCTS reward with a **final-answer-only reward** and trained a variant (**64 iters(ORL)**). The notably lower performance of this variant confirms that **the SPRE reward design yields greater gains**.
>
> [1]. Luo, Liangchen, et al. "Improve mathematical reasoning in language models by automated process supervision." arXiv preprint arXiv:2406.06592 (2024).
>
> ## **2. Stepwise Scaling on Cost and Accuracy**
>
> > Moreover, the datasets used in their experiments involve no more than five steps per episode. It remains unclear how the computational cost and accuracy would scale as the number of steps increases.
>
> > How do computational cost and accuracy vary with the number of steps in a sample? Since the current datasets contain at most five steps, it would be valuable to understand how the method scales with longer reasoning trajectories.
>
>
> We conduct a step-wise cost and accuracy analysis across both single-hop (PopQA) and multi-hop datasets (HotpotQA, 2WikiMultiHopQA). As shown in the table above:
>
> | Step | PopQA EM | PopQA Total Time(s) | HotpotQA EM | HotpotQA Total Time(s) | 2Wiki EM | 2Wiki Total Time(s) |
> | ---- | -------- | ---------------- | ----------- | ------------------- | -------- | ---------------- |
> | 1    | 15.6     | 24             | 15.1        | 28                | 26.4     | 29             |
> | 2    | 35.8     | 51             | 29.6        | 61               | 30.1     | 64             |
> | 3    | 39.8     | 66             | 35.0        | 84               | 39.8     | 81             |
> | 4    | 40.2     | 75            | 37.1        | 105              | 41.5     | 100            |
> | 5    | 40.5     | 80            | 38.3        | 121             | 42.1     | 110            |
>
> ### **2.1 Key findings**
> We observe **consistent performance gains** across both single-hop and multi-hop datasets as the number of reasoning steps increases. For single-hop QA, most gains are achieved in early steps, and further steps bring marginal improvements. Despite longer total inference times, our method mitigates latency by **executing each reasoning step in parallel** across all queries using a batched vLLM engine.
>
>
> ## **3. Computational Cost Comparison**
> > What are the computational costs of the different algorithms listed in Tables 2 and 3? A comparison in terms of runtime or sample efficiency would help contextualize the performance gains.
>
> Thank you for the helpful suggestion. We have added a runtime and efficiency comparison using 1,000 queries from the 2WikiMultihopQA dataset. The runtime analysis are shown as below:
>
> | Method               | Time (s) | Avg. Tokens | Avg. Retrievals | Avg. EM  | Avg. F1  |
> | -------------------- | -------- | ----------- | --------------- | -------- | -------- |
> | Naïve Generation     | 2.7      | 54          | 0               | 11.5     | 19.4     |
> | Standard RAG         | 29       | 763         | 1.0             | 24.3     | 32.0     |
> | FLARE                | 42       | 763         | 5.0             | 15.3     | 21.9     |
> | AdaptiveRAG          | 682      | 3243        | 4.0             | 23.0     | 31.2     |
> | Iter-RetGen          | 103      | 2136        | 3.0             | 25.4     | 33.3     |
> | IRCoT                | 732      | 4379        | 4.8             | 22.6     | 30.7     |
> | RECOMP               | 31       | 807         | 1.0             | 26.9     | 34.2     |
> | LongLLMLingua        | 31       | 823         | 1.0             | 26.8     | 34.2     |
> | Selective-Context    | 32       | 834         | 1.0             | 19.2     | 27.0     |
> | AutoRAG              | 130      | 2563        | 4.2             | 29.5     | 36.9     |
> | Search-R1            | 89       | 1683        | 2.8             | 32.8     | 40.7     |
> | Search-O1            | 115      | 2432        | 4.4             | 22.2     | 31.96    |
> | **ReasonRAG (Ours)** | **110**  | **2304**    | **3.8**         | **34.4** | **42.3** |
> (Note that all methods use the vllm inference engine.)
>
> ### **3.1 Key findings**
> ReasonRAG demonstrates the best overall performance (Avg. EM/F1) among all evaluated methods, while maintaining lower latency than baselines with comparable or higher accuracy.
>
>
> ## **4. Training on more LLM variant: Qwen2.5-3B**
>
> > Have you tried using different base models? It would be helpful to know whether the observed improvements are consistent across models of varying sizes or architectures.
>
> Thank you for the valuable question. We conducted additional experiments with all baselines and our method implemented using **Qwen2.5-3B-Instruct** as the base model. The results are presented below:
>
> | Method             | PopQA EM | PopQA F1 | HotpotQA EM | HotpotQA F1 | 2WikiMulti EM | 2WikiMulti F1 | Bamboogle EM | Bamboogle F1 | MuSiQue EM | MuSiQue F1 | Avg. EM  | Avg. F1  |
> | ------------------ | -------- | -------- | ----------- | ----------- | ------------- | ------------- | ------------ | ------------ | ---------- | ---------- | -------- | -------- |
> | Naïve Generation   | 10.8     | 12.6     | 14.9        | 16.7        | 10.0          | 15.3          | 2.4          | 14.9         | 2.0        | 4.6        | 8.0      | 12.8     |
> | Standard RAG       | 32.4     | 36.6     | 25.6        | 33.6        | 22.0          | 27.5          | 4.0          | 12.0         | 3.5        | 9.3        | 17.5     | 23.8     |
> | Search-O1          | 21.6     | 25.9     | 15.1        | 25.0        | 14.7          | 22.7          | 6.4          | 14.0         | 2.5        | 8.0        | 12.1     | 19.1     |
> | Search-R1 (90k)    | 30.0     | 35.0     | 30.4        | 36.0        | 26.0          | 31.8          | 12.5         | 19.0         | 7.6        | 13.0       | 21.3     | 27.0     |
> | **ReasonRAG (5k)** | **32.9** | **36.8** | **30.0**    | **41.3**    | **26.6**      | **32.1**      | **13.6**     | **23.7**     | *6.9*      | **13.2**   | **22.0** | **29.4** |
>
> ### **4.1 Key findings**
> These results demonstrate that ReasonRAG generalizes well to smaller models, maintaining strong performance across tasks and outperforming several strong baselines. Thank you for your valuable feedback.
>
> We hope these responses address your concerns comprehensively. Feel free to let us know if you have any additional questions, and we are more than happy to provide further clarification on any aspect of our work.
>
> Best regards,
>
> The Authors

---

> ### Author Response · Authors · 2025-08-05
> **Kind Reminder: Rebuttal Submitted**
>
> Dear Reviewer WaUS,
>
> We hope this message finds you well. We would like to gently follow up regarding our submitted rebuttal. In our response, we have made every effort to thoroughly address the reviewers’ comments, clarify our contributions, and thoughtfully respond to the concerns raised.
>
> If time permits, we would be truly grateful if you could kindly take another look at our submission and consider whether our clarifications may warrant a reevaluation. Your insights are highly valued, and we are more than happy to provide any further information if needed.
>
> Thank you sincerely for your time and consideration.

---

> > ### Author Response · Authors · 2025-08-07
> > **Re: Kind Reminder: Rebuttal Submitted**
> >
> > Dear Reviewer WaUS,
> >
> > As the rebuttal deadline is approaching, we would like to kindly follow up on our earlier message regarding our submitted response. We have done our best to address all comments with care and detail.
> >
> > If there are any remaining concerns or questions, we would be very glad to clarify further. Your feedback is truly appreciated, and we remain fully available to assist if needed.
> >
> > Thank you again for your time and consideration.
> >
> > Best regards,
> >
> > Authors

---

> > > ### Author Response · Authors · 2025-08-09
> > > **Re: Re: Kind Reminder: Rebuttal Submitted**
> > >
> > > Dear Reviewer WaUS,
> > >
> > > As the discussion phase is coming to an end soon, we wanted to reach out once more regarding our submitted rebuttal. We have made every effort to address your comments thoroughly, and we are happy to expand on any points that may still need clarification.
> > >
> > > Your feedback would be invaluable in ensuring all concerns are resolved before the deadline. We greatly appreciate the time and attention you have given to reviewing our work.
> > >
> > > Thank you again, and we look forward to hearing from you.
> > >
> > > Warm regards,
> > >
> > > Authors

---

### Official Review · Reviewer_ATvT · 2025-07-03

**Clarity:** 3
**Significance:** 2
**Originality:** 2
**Rating:** 4
**Confidence:** 4

**Summary:**

This paper addresses the limitations of outcome supervised RL for RAG by introducing a process supervised framework called ReasonRAG. Unlike prior methods that only reward the final answer’s correctness, ReasonRAG leverages fine grained, stepwise rewards constructed via their Shortest Path Reward Estimation algorithm (SPRE). They also employ Monte Carlo Tree Search to explore and annotate intermediate reasoning trajectories. The resulting dataset (5 k questions, 13 k preference pairs) is called RAG-Proguide which is used to optimize the agentic RAG policy via DPO. Experiments were done on five QA benchmarks  and show that ReasonRAG trained on only 5k queries, outperforms Search-R1 baseline (90 k queries) in both EM and F1 while using less GPU compute

**Questions:**

1. To prove the generality of the Method it will be interesting if the authors can perform experiment using a simple llama-3.1 or 3.3 model against search-o1 and search-r1 on 2 unseen datasets only. This will strength the papers claims. And take care of another baseline (i.e search-o1) .
2. Giving the run time cost, speed , searches done and tokens for the inference will significantly help the paper.

**Ethical Concerns:**

["NO or VERY MINOR ethics concerns only"]

**Final Justification:**

1. experiments included a smaller LLM to show generatily
2. The inference table actually shows that ReasonRAG is almost as efficient as other baselines while being far better in terms of EM and F1 scores. This gives the confidence that the method is really good compared to the baselines.
3. Overall rebuttal felt very convincing.

**Limitations:**

Yes

**Paper Formatting Concerns:**

Page Limit of 9 is not followed strictly. A small part of the conclusion section is present in the page-10.

**Quality:**

3

**Strengths And Weaknesses:**

Strengths:
1. The introduction of SPRE to estimate process‐level rewards by penalizing unnecessarily long reasoning paths is a novel contribution that directly addresses sparse and delayed rewards in RL for RAG.
2. Mixing MCTS and SPRE to label data by exploring and minimizing the cost of human annotation is an interesting approach.
3. Proposed method is definitely compute efficient compared to Search-R1 in terms of GPU hours and the statistical significance test in the  table 2 is convincing.

Weaknesses:
1. Experiments use a Qwen2.5-7B-Instruct model only. It is uncertain whether ReasonRAG’s benefits translate to smaller or less capable LLMs.
2. Although MCTS is used offline for annotation, the inference algorithm still involves iterative reasoning loops which may incur latency in real‐time applications. An analysis of runtime cost per query would strengthen practicality claims.
3. While the authors claimed that they used 2 OOD datasets (unseen) it is not very convincing see the gains are marginal over the search-r1 in those 2 datasets (Bamboogle, MusiQue). And No recent baselines like search-O1.
4. The base & SFT performance of the ReasonRAG is closer or same as standard RAG (table-3 row 1-2 and table 2 zero shot section) this requires further analysis on what exactly is happening.

---

> ### Author Rebuttal · Authors · 2025-07-31
>
> Dear Reviewer ATvT,
>
> We sincerely thank you for your thoughtful review and valuable feedback on our work. Below, we address each of your concerns with additional experimental evidence and clarifications.
>
> ## **1. Training on more LLM variant: Qwen2.5-3B**
>
> > Experiments use a Qwen2.5-7B-Instruct model only. It is uncertain whether ReasonRAG’s benefits translate to smaller or less capable LLMs.
>
> > To prove the generality of the Method it will be interesting if the authors can perform experiment using a simple llama-3.1 or 3.3 model against search-o1 and search-r1 on 2 unseen datasets only.
>
> Thank you for the valuable comment. We conducted additional experiments by re-implementing all baselines using **Qwen2.5-3B-Instruct** as the backbone. The results are reported below:
>
> | Method | PopQA EM | PopQA F1 | HotpotQA EM | HotpotQA F1 | 2WikiMulti EM | 2WikiMulti F1 | Bamboogle EM | Bamboogle F1 | MuSiQue EM | MuSiQue F1 | Avg. EM | Avg. F1 |
> |---------------------|----------|----------|--------------|--------------|----------------|----------------|----------------|----------------|--------------|--------------|---------|---------|
> | Naïve Generation | 10.8| 12.6| 14.9 | 16.7 | 10.0 | 15.3 | 2.4 | 14.9 | 2.0| 4.6 | 8.0| 12.8 |
> | Standard RAG | 32.4| 36.6| 25.6 | 33.6 | 22.0 | 27.5 | 4.0 | 12.0 | 3.5| 9.3 | 17.5 | 23.8 |
> | Search-O1 | 21.6| 25.9| 15.1 | 25.0 | 14.7 | 22.7 | 6.4 | 14.0 | 2.5| 8.0 | 12.1 | 19.1 |
> | Search-R1 (90k)| 30.0| 35.0| 30.4 | 36.0 | 26.0 | 31.8 | 12.5 | 19.0 | 7.6| 13.0| 21.3 | 27.0 |
> | **ReasonRAG (5k)** | **32.9** | **36.8** | **30.0**| **41.3**| **26.6** | **32.1** | **13.6*** | **23.7*** | _6.9_ | **13.2** | **22.0** | **29.42** |
>
> ### **1.1 Key findings:**
> (1) **​ReasonRAG remains effective when applied to smaller language models**​, achieving strong performance with Qwen2.5-3B-Instruct and outperforming baseline methods across most datasets.
> (2) **The performance gains are especially notable on multi-hop QA tasks** such as HotpotQA and 2WikiMultihopQA, suggesting that ReasonRAG is particularly effective in complex reasoning scenarios.
>
> ## **2. Inference Efficiency vs. Efficacy**
>
> > Although MCTS is used offline for annotation, the inference algorithm still involves iterative reasoning loops which may incur latency in real-time applications. An analysis of runtime cost per query would strengthen practicality claims.
>
> > Giving the run time cost, speed , searches done and tokens for the inference will significantly help the paper.
>
> Thank you for your suggestion about runtime cost analysis. We have added a runtime comparison across methods based on 1,000 queries from 2WikiMultihopqa dataset. All experiments use the Qwen2.5-7B-Instruct model with the vLLM inference engine for efficient batched decoding, and baseline implementations follow the open-source FlashRAG framework:
>
> | Method               | Time (s) | Avg. Tokens | Avg. Retrievals | Avg. EM  | Avg. F1  |
> | -------------------- | -------- | ----------- | --------------- | -------- | -------- |
> | Naïve Generation     | 2.7      | 54          | 0               | 11.5     | 19.4     |
> | Standard RAG         | 29       | 763         | 1.0             | 24.3     | 32.0     |
> | FLARE                | 42       | 763         | 5.0             | 15.3     | 21.9     |
> | AdaptiveRAG          | 682      | 3243        | 4.0             | 23.0     | 31.2     |
> | Iter-RetGen          | 103      | 2136        | 3.0             | 25.4     | 33.3     |
> | IRCoT                | 732      | 4379        | 4.8             | 22.6     | 30.7     |
> | RECOMP               | 31       | 807         | 1.0             | 26.9     | 34.2     |
> | LongLLMLingua        | 31       | 823         | 1.0             | 26.8     | 34.2     |
> | Selective-Context    | 32       | 834         | 1.0             | 19.2     | 27.0     |
> | AutoRAG              | 130      | 2563        | 4.2             | 29.5     | 36.9     |
> | Search-R1            | 89       | 1683        | 2.8             | 32.8     | 40.7     |
> | Search-O1            | 115      | 2432        | 4.4             | 22.2     | 31.96    |
> | **ReasonRAG (Ours)** | **110**  | **2304**    | **3.8**         | **34.4** | **42.3** |
> (Note that all methods use the vllm inference engine.)
>
> ### **2.1 Key findings:**
> ReasonRAG demonstrates the best overall performance (Avg. EM/F1) among all evaluated methods, while maintaining lower latency than baselines with comparable or higher accuracy.
>
> ## **3. OOD Generation**
> > While the authors claimed that they used 2 OOD datasets (unseen) it is not very convincing see the gains are marginal over the search-r1 in those 2 datasets (Bamboogle, MusiQue).
>
> Thank you for your valuable feedback. We agree that gains on OOD datasets are an important indicator of generalization. While the performance improvements on *Bamboogle* and ​*MusiQue* over Search-R1 are relatively smaller than on in-domain datasets, it is worth noting that ReasonRAG is trained with only ​**5k queries**​, whereas Search-R1 is trained on ​**90k queries**​. Despite this 18× smaller training set, ReasonRAG achieves ​**stronger overall performance**​, including notable improvements on Bamboogle and competitive results on MusiQue, demonstrating both generalization and data efficiency​.
>
> ## **4. Results of Additional Baseline: Search-O1**
> > And No recent baselines like search-O1.
>
> > Q1: Take care of another baseline (i.e search-o1).
>
> We have included the performance of ​**Search-O1**​ as shown in the table below:
>
> | Method | PopQA EM | PopQA F1 | HotpotQA EM | HotpotQA F1 | 2Wiki EM | 2Wiki F1 | Bamboogle EM | Bamboogle F1 | Musique EM | Musique F1 | Avg. EM | Avg. F1 |
> |--------------------|----------|----------|-------------|-------------|----------|----------|---------------|----------------|------------|-------------|---------|---------|
> | Search-O1| 33.2| 40.3| 24.8 | 38.1 | 16.4| 27.1| 30.4| 40.6 | 6.3 | 13.7 | 22.2 | 31.96 |
> | AutoRAG (10k) | 38.6| 44.1| 33.3 | 43.7 | 39.5| 46.1| 24.8| 32.2 | 11.3 | 18.3 | 29.5 | 36.9 |
> | Search-R1 (90k) | 39.7| 44.8| 37.0 | 47.0 | 41.4| 48.0| 32.0| 43.8 | **14.6** | 19.9 | 32.8 | 40.7 |
> | **ReasonRAG (5k)** | **41.5** | **46.2** | **38.4** | **48.9** | **43.6** | **50.4** | **36.0** | **45.5** | 12.8 | **20.6** | **34.4**| **42.3**|
>
> ### **4.1 Key findings:**
> ReasonRAG outperforms both Search-R1 and Search-O1 across nearly all datasets and metrics, further validating the effectiveness and robustness of our approach.
>
>
> ## **5. Clarification of ReasonRAG Base and SFT**
> > The base & SFT performance of the ReasonRAG is closer or same as standard RAG (table-3 row 1-2 and table 2 zero shot section) this requires further analysis on what exactly is happening.
>
> Thank you for your insightful question. To clarify the differences, we present the comparison results below:
>
> | Method | PopQA EM | PopQA F1 | HotpotQA EM | HotpotQA F1 | 2WikiMulti EM | 2WikiMulti F1 | Bamboogle EM | Bamboogle F1 | MuSiQue EM| MuSiQue F1| Avg. EM | Avg. F1 |
> | ------------------------------ | ---------------- | ---------------- | ---------------- | ---------------- | ---------------- | ---------------- | ---------------- | ---------------- | ---------------- | ---------------- | ---------------- | ---------------- |
> | Standard RAG | 38.4 | 44.7 | 29.3 | 39.9 | 29.4 | 36.3 | 17.6 | 24.1 | 6.7 | 15.1 | 24.3 | 32.0 |
> | ReasonRAG (Base) | 35.6 | 42.7 | 23.7 | 38.2 | 15.2 | 28.9 | 28.0 | 38.7 | 7.7 | 15.4 | 22.0 | 32.8 |
> | ReasonRAG (SFT) | 31.6 | 37.4 | 26.8 | 38.7 | 35.1 | 40.9 | 17.6 | 27.3 | 8.6 | 15.5 | 23.9 | 32.0 |
> | **ReasonRAG (PRL)** | **41.5** | **46.2** | **38.4** | **48.9** | **43.6** | **50.4** | **36.0** | **45.5** | **12.8** | **20.6** | **34.5** | **42.3** |
>
> **Why ReasonRAG-Base falls short**: Unlike standard RAG which passively uses retrieved documents, ReasonRAG requires active agentic capabilities—planning retrieval strategies, issuing targeted queries, and orchestrating multi-step reasoning. The base model lacks these learned behaviors, making it more similar to untrained agentic baselines like Search-O1 than to standard RAG, resulting in degraded performance across tasks.
> **Why ReasonRAG-SFT fails to generalize**: SFT training on high-reward trajectories from PopQA, HotpotQA, and 2WikiMultihopQA creates task-specific overfitting. While it improves performance on seen datasets (HotpotQA, 2WikiMultihopQA), it fails on unseen tasks like Bamboogle and MuSiQue. This aligns with recent findings that "SFT memorizes, RL generalizes" [1]—the model learns surface patterns rather than transferable reasoning strategies.
> **Why ReasonRAG-PRL succeeds**: PRL leverages both positive and negative feedback with fine-grained reward signals based on model preferences, encouraging flexible reasoning behaviors rather than memorized patterns. This approach enables the model to develop generalizable agentic strategies that transfer across diverse QA tasks, achieving consistent improvements (42.3% avg F1) compared to both base (32.8%) and SFT (32.0%) variants.
>
>
> [1] Chu, Tianzhe, et al. "Sft memorizes, rl generalizes: A comparative study of foundation model post-training." arXiv preprint arXiv:2501.17161 (2025)
>
> We hope these responses address your concerns comprehensively. Feel free to let us know if you have any additional questions, and we are more than happy to provide further clarification on any aspect of our work.
>
> Best regards,
>
> The Authors

---

> > ### Comment · Reviewer_ATvT · 2025-08-09
> >
> > Thanks for the detailed address to all the weaknesses, overall i am fully satisfied of the responses and would urge the authors to include these details in the camera ready and it would be a great help to the opensource community to have codebase public once the paper is published.
> >
> > 1. Just a minor concern left for me is that you didn't include the total training time + compute comparision of various baselines in your Point 2. And this can help us understand the overall training efficiency too.
> > 2. For the base ReasonRAG to be lesser than standard RAG , the explanation is not very convincing somehow. The general expectation is that ReasonRAG with all tools even without finetuning should be able to do better in all the benchmarks and the results show here that this is not consistent. would appreciate a little more analysis on why this is not the case.
> >
> > I have improved my scores accordingly and would be waiting for the further explanations about 1and 2.

---

> ### Author Response · Authors · 2025-08-05
> **Kind Reminder: Rebuttal Submitted**
>
> Dear Reviewer ATvT,
>
> We would like to kindly remind you that we have submitted our rebuttal. We have carefully addressed the reviewers’ comments and provided detailed responses to clarify our contributions and resolve the raised concerns.
>
> If possible, we would greatly appreciate it if you could take a moment to review our responses and consider the possibility of raising the score of our submission. Your feedback is very valuable to us, and we are happy to provide any further clarification if needed.
>
> Thank you very much for your time and consideration

---

> > ### Author Response · Authors · 2025-08-07
> > **Re: Kind Reminder: Rebuttal Submitted**
> >
> > Dear Reviewer ATvT,
> >
> > We are writing to gently follow up on our previous email regarding the rebuttal for our submission. As the rebuttal deadline is approaching, we would like to kindly remind you that we have already submitted our response and have done our best to address all reviewer comments in detail.
> >
> > If there are any remaining questions or points of clarification you would like us to elaborate on, we would be very happy to assist.
> >
> > Thank you again for your time and valuable feedback. We sincerely appreciate your efforts and look forward to hearing from you.
> >
> > Warm regards,
> > Authors

---

> > > ### Author Response · Authors · 2025-08-09
> > > **Re: Re: Kind Reminder: Rebuttal Submitted**
> > >
> > > Dear Reviewer ATvT,
> > >
> > > As the rebuttal discussion deadline is now very close, we would like to kindly follow up once again regarding our previous response to your comments.
> > >
> > > We have carefully addressed all points raised in the initial review and would be very happy to provide any additional clarifications or details you may need before the discussion period closes.
> > >
> > > Your feedback is extremely valuable for improving our work, and we truly appreciate the time and effort you have dedicated to reviewing our submission.
> > >
> > > Thank you again for your consideration, and we look forward to your reply.
> > >
> > > Warm regards,
> > > Authors

---

### Note · Authors · 2025-08-12

Dear Area Chair and Reviewers,

We sincerely thank all reviewers for their constructive feedback and insights. We greatly appreciate the recognition of our contributions and are encouraged by the positive comments highlighting the strengths of our work:

* **Novel process-level reward design**: Introduction of *Shortest Path Reward Estimation* (SPRE) to address sparse and delayed rewards in RL for RAG, combined with MCTS to explore and annotate reasoning trajectories efficiently. (Reviewers ATvT, WaUS, 2RBG)
* **Strong empirical performance with data efficiency**: Outperforms the large-scale outcome-based baseline (Search-R1) on five QA benchmarks using only 5k queries. (Reviewers ATvT, WaUS, aFno)
* **Comprehensive and convincing evaluation**: Experiments cover diverse benchmarks, include significance tests, and analyze exploration diversity and iteration-count distributions. (Reviewers ATvT, 2RBG)

A major concern shared by several reviewers is that the experiments use a **Qwen2.5-7B-Instruct model only**, making it uncertain whether ReasonRAG’s benefits translate to smaller or less capable LLMs.
Another recurring concern is the **lack of detailed reasoning time and efficiency comparison** with baselines.
In addition, reviewers noted the need for **clearer ablation experiments to identify where ReasonRAG’s gains come from**.

To address these concerns, we have:

* Added experiments with **Qwen2.5-3B-Instruct** to evaluate the method’s effectiveness on smaller-scale LLMs.
* Included detailed **inference cost and efficiency analyses** comparing ReasonRAG with baselines.
* Conducted comprehensive **ablation studies** isolating the contributions of reward design, MCTS, and training strategy.

We are committed to addressing all concerns and polishing our paper to make it more robust and accessible to the community. Specifically, in the revised version we will:

1. Update results with the new Qwen2.5-3B experiments.
2. Incorporate the inference cost analysis into the main results section.
3. Include the expanded ablation results with detailed discussions.

Finally, we sincerely appreciate the encouraging feedback on the potential impact of our work. Several reviewers noted that our approach could inspire further research in the RAG community and beyond. We emphasize that our contributions go beyond RAG, offering broader applicability and insights, and we believe our work is worthy of publication to foster further discussion and exploration.

---

### Decision · Program_Chairs · 2025-09-17

**Decision:**

Accept (poster)

**Comment:**

The paper introduces ReasonRAG, a novel framework for agentic Retrieval-Augmented Generation (RAG) that leverages process-supervised reinforcement learning (RL) to enhance large language models’ (LLMs) capabilities in complex, multi-step reasoning tasks. The authors propose RAG-ProGUIDE, a high-quality dataset providing fine-grained, process-level rewards for query generation, evidence extraction, and answer synthesis, constructed using Monte Carlo Tree Search (MCTS) and Shortest Path Reward Estimation (SPRE). Experimental results demonstrate that ReasonRAG outperforms existing methods like Search-R1 and traditional RAG systems across five benchmark datasets (PopQA, HotpotQA, 2WikiMultiHopQA, Bamboogle, and MusiQue), achieving superior performance with only 5k training instances compared to 90k for Search-R1. The paper claims that process-level supervision, enabled by SPRE and MCTS, addresses limitations of outcome-based RL, such as sparse rewards and low exploration efficiency, leading to improved generalization and data efficiency.

Reviewers mention several weakness of the paper:

MCTS Novelty: MCTS-based RAG systems (e.g., AirRAG) exist. The authors clarified their unique use of MCTS for offline training data generation.

Initial Ablation Gaps: The original submission lacked detailed component-wise ablations, addressed in the rebuttal with clear experiments.

Inference Latency: Concerns about iterative reasoning loops were mitigated with runtime comparisons showing competitive latency (110s for 1,000 queries).

Penalty Coefficient Analysis: Limited initial analysis of (\alpha) was resolved with a sensitivity study showing performance impacts.

Out-of-Distribution Gains: Marginal improvements on Bamboogle and MusiQue were noted, but justified by the 18× smaller training set.

During rebuttal, the authors have addressed most reviewer concerns through additional experiments and clarifications. Two reviewers didn't engage in the rebuttal. Taking this into account, I think this paper's contribution and interesting topic outweight the weakness. Authors shall include all the new experiments  (e.g., Qwen2.5-3B, (\alpha) analysis).